# Improving Progressive Generation with Decomposable Flow Matching

**Moayed Haji-Ali**[*,†,1,2]     **Willi Menapace**[*,2]     **Ivan Skorokhodov**[2]     **Arpit Sahni**[2]

**Sergey Tulyakov**[2]     **Vicente Ordonez**[1]     **Aliaksandr Siarohin**[2]

[1]Rice University        [2]Snap Inc

Project Webpage: https://snap-research.github.io/dfm

## Abstract

Generating high-dimensional visual modalities is a computationally intensive task. A common solution is *progressive generation*, where the outputs are synthesized in a coarse-to-fine spectral autoregressive manner. While diffusion models benefit from the coarse-to-fine nature of denoising, explicit multi-stage architectures are rarely adopted. These architectures have increased the complexity of the overall approach, introducing the need for a custom diffusion formulation, decomposition-dependent stage transitions, ad-hoc samplers, or a model cascade. Our contribution, Decomposable Flow Matching (DFM), is a simple and effective framework for the progressive generation of visual media. DFM applies Flow Matching independently at each level of a user-defined multi-scale representation (such as Laplacian pyramid). As shown by our experiments, our approach improves visual quality for both images and videos, featuring superior results compared to prior multi-stage frameworks. On Imagenet-1k 512px, DFM achieves 35.2% improvements in Frechet DINOv2 Distance (FDD) scores over the base architecture and 26.4% over the best-performing baseline, under the same training compute. When applied to finetuning of large models, such as FLUX, DFM shows faster convergence speed to the training distribution. Crucially, all these advantages are achieved with a single model, architectural simplicity, and minimal modifications to existing training pipelines.

## 1 Introduction

The high dimensionality of images and videos poses a challenge for generative modeling. An approach to make the problem more tractable involves decomposing the generative task into a sequence of simpler sub-problems. An increasing number of generative models adopt this practice by progressively modeling visual signals of increasing resolution [1, 11, 16, 18, 41, 42]. Autoregressive generative models, for instance, have demonstrated that switching from a next-token prediction pattern in a 1D sequence of flattened tokens [4, 6, 24, 31, 35, 49, 50] to a next-scale [42] or next-frequency [16] prediction pattern, yields substantial performance improvements. Recently, the success of diffusion models has been connected to a form of spectral autoregression enforced by the diffusion process which erases details progressively from the coarsest to the finest [30, 36, 38].

---

[*]Equal Contribution.
[†]Work partially done during an internship at Snap Inc.

39th Conference on Neural Information Processing Systems (NeurIPS 2025).

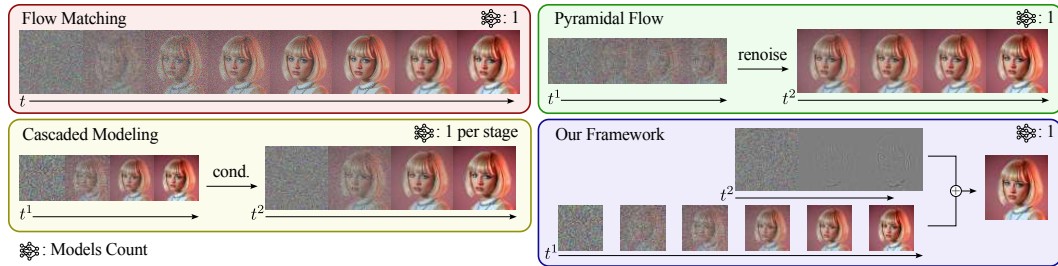

Figure 1: Comparison of different diffusion frameworks. Cascaded models employ a separate model per stage. Pyramidal Flow [18] progressively increases resolution through careful upsampling and renoising operations. Our framework models multiple stages independently from each other within the same model, enabling progressive generation with minimal modifications to Flow Matching [27]

A variety of diffusion frameworks [1, 9, 13, 14, 18, 37, 41] have reformulated the diffusion process to enforce spectral autoregression, an approach we refer to as *progressive modeling*. This is typically realized by restructuring the diffusion process to include transition points where the signal resolution is altered, thereby defining a sequence of stages of progressively increasing resolution. Key progressive modeling paradigms are illustrated in Figure 1, where cascaded models [13, 14, 37] and Pyramidal Flow [18] are exemplified and contrasted with vanilla Flow Matching [26, 27]. However, while these frameworks demonstrate the feasibility of the progressive paradigm, they often operate directly in pixel space [1, 41], depend on multiple models [13, 14, 37, 41], or deviate from the original formulation of the diffusion process [1, 9, 18, 41]. Such deviations introduce complexities in managing inter-stage transitions, requiring careful consideration of operations such as resizing [1, 18, 41], renoising [1, 18, 41] or blurring [41] and, in some instances, require specialized samplers [9, 41].

To address these challenges, we propose Decomposable Flow Matching (DFM), a progressive modeling framework based on a simple extension of Flow Matching, depicted in Figure 2. At its core, a user-defined decomposition function is applied to the input, producing a multiscale input representation that defines each *stage* of the generation, and a Flow Matching process is independently applied to each stage, utilizing a set of per-stage flow timesteps. During inference, a standard sampler operates separately on each stage of the representation guided by a user-defined sampler schedule that dictates the progression of flow timesteps per each stage. Timesteps are evolved sequentially, starting from the coarsest stage; progression to a subsequent stage only begins once the preceding stage has reached a predefined threshold. During training, progressive generation is simulated by selecting a stage, sampling a noise level for it, and injecting the maximal noise level into subsequent stages of the representation and a small noise level to preceding ones. These capabilities are realized without the need for expert models, ad-hoc diffusion processes, or specialized samplers, all while offering the flexibility of user-defined input decomposition.

We conduct a thorough analysis of the design space for DFM, providing insights into its behavior across different training and sampling strategies, and model architectures. Based on these findings, we instantiate DFM and evaluate its performance on established image and video generation benchmarks, specifically ImageNet-1K [5] and Kinetics-700 [3]. Our results demonstrate uniform improvements over Flow Matching [27], cascaded models, and Pyramidal Flow [18]. Furthermore, we apply DFM to the finetuning of FLUX [23], demonstrating its efficacy in enhancing the performance of large-scale generative models. Compared with standard full finetuning, applying DFM for the same amount of finetuning iterations yields a 29.7% reduction in FID and 3.7% increase in CLIP score.

In summary, our work introduces Decomposable Flow Matching, a novel progressive generation framework that offers several key advantages: (i) It presents a simple formulation rooted in Flow Matching; (ii) it is agnostic to the choice of decompositions; (iii) it does not need multiple models or per-stage training; (iv) it includes a comprehensive analysis of critical training and sampling hyperparameters; (v) it outperforms existing state-of-the-art progressive generation frameworks on ImageNet-1K [5] and Kinetics-700 [3], and enhances FLUX [23] finetuning performance.

## 2 Related Work

### 2.1 Progressive Generation in Autoregressive Models

Autoregressive models for visual generation [4, 6, 24, 31, 35, 49, 50] formulate the generation of visual data as a next-token prediction problem. This is typically achieved by first obtaining a

---

**Algorithm 1** Sampling procedure of training timesteps across multi-scale stages

---

1: **procedure** SAMPLETRAININGTIMESTEPS($S$)
    **Input:** $S$                                                      ▷ total number of scales
    **Output:** $(t^1, \ldots, t^S)$
2:     $Q \leftarrow$ SAMPLEDISCRETEDISTRIBUTION($p_q$)         ▷ stage index corresponding to #scales
3:     $t^Q \leftarrow$ SAMPLELOGITNORMAL($m = 0$)
4:     **for** $s = 1$ **to** $Q - 1$ **do**
5:         $t^s \leftarrow$ SAMPLELOGITNORMAL($m = 1.5$)
6:     **end for**
7:     **for** $s = Q + 1$ **to** $S$ **do**
8:         $t^s \leftarrow 0$
9:     **end for**
10:    **return** $(t^1, \ldots, t^S)$
11: **end procedure**

---

discrete input representation, which is then flattened into a 1D sequence to enable autoregressive modeling. However, recent work [16, 42] has demonstrated that flattening undermines the intrinsic multidimensional structure of visual data. VAR [42] addresses this limitation by modeling image generation as a spectral autoregressive process. A multiscale tokenizer generates discrete tokens at different resolutions, and the model predicts each scale conditioned on the previous ones, yielding substantial improvements over baselines. NFIG [16] adopts a similar approach where frequency decomposition is used to define different scales that are predicted autoregressively. In our work, we explore spectral autoregression in the context of diffusion models.

## 2.2 Progressive Generation in Diffusion Models

Diffusion models have achieved state-of-the-art results across various visual generation tasks [7, 23, 28, 46]. Their effectiveness has been linked to an implicit form of *spectral autoregression*, whereby the diffusion process erases fine-grained details in a coarse-to-fine fashion [30, 36, 38].

Several *progressive generation* frameworks made spectral autoregression an explicit part of their design, generating their outputs stage-by-stage. Cascaded diffusion models [13, 14, 37] generate low-resolution outputs in an initial stage, followed by upsampling stages conditioned on previous outputs. A separate model is used for each stage. Other methods redefine the generative process to incorporate progressive signal transformations. f-DM [9] and RDM [41] introduce stages via downsampling. MDM [10] simultaneously denoises multiple resolutions. Edify Image [1] introduces frequency-specific attenuation in the diffusion process. Pyramidal Flow Matching [18] operates in the latent space over a hierarchy of increasing resolutions, combining upsampling and re-noising operations. Despite these advances, existing approaches share limitations. They often require complex stage-transition mechanisms [1, 9, 18, 41] or tailored samplers [9, 41], are tightly coupled to specific decomposition strategies [1, 9, 18, 41], and primarily focus on pixel-space generation [1, 10, 41]. In contrast, we propose a simple and generic framework for progressive generation in the latent space that supports user-defined decompositions without altering the core architecture.

## 2.3 Autoencoders with Spectral Latent Structure

Coarse-to-fine or *spectral autoregressive* modeling benefits from an autoencoder with a well-structured spectral representation where low-, mid-, and high-frequency components in the RGB space correspond to the respective low-, mid-, and high-frequency components in the latent representation. Modern image and video autoencoders typically lack such spectral structure [38], so recent works fine-tune them to enforce the desired spectral properties through scale equivariance regularization [17, 22, 38]. Another viable strategy is to train a Spectral Image Tokenizer [8]-like model from scratch, which encodes a wavelet-decomposed image representation into a causally ordered sequence, yielding spectrally structured latents by design. Our framework is agnostic to the choice of the autoencoder but benefits from such spectral latent structure, and we opt for the first strategy of fine-tuning with scale equivariance [22, 38] to stay closer to existing diffusion pipelines.

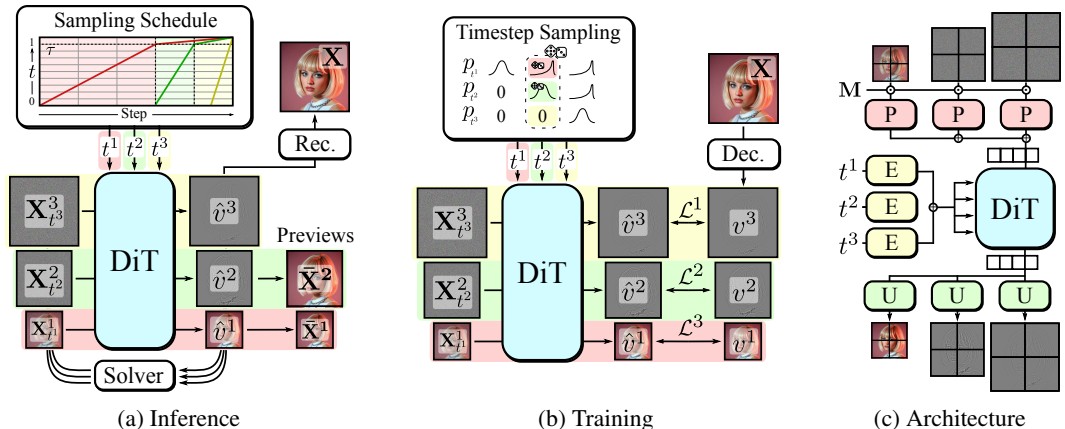

(a) Inference       (b) Training       (c) Architecture

Figure 2: Overview of Decomposable Flow Matching (a) Sampling procedure. A sampling schedule is defined that dictates the timestep progression of each stage. The model predicts velocities for each input scale and a vanilla sampler denoises each scale according to its schedule. Intermediate stages can decoded during inference to provide a low-resolution preview of the generation. (b) Training procedure. A stage index is sampled from a discrete probability distribution and defines the main stage. Timesteps for each stage are sampled accordingly. The model minimizes velocity prediction errors for each scale. (c) DiT [33] architecture for progressive generation. Patchification layers and time embedders are replicated for each scale.

## 3 Method

This section details our progressive generation framework, summarized in Figure 2. The method is based on Flow Matching [26, 27], which is introduced in Section 3.1. A Laplacian decomposition strategy, detailed in Section 3.2, is exemplified as a way of producing a multiscale input representation amenable to stage-by-stage progressive modeling. In Section 3.3, the Flow Matching framework is extended to account for the multiple stages by introducing per-stage flow timesteps, which are leveraged to instantiate the progressive generation scheme by denoising stages one-by-one, starting from the coarsest. Finally, the DiT [33] architecture is extended in Section 3.4 to support the proposed generation scheme by introducing per-scale patchification and time embedding layers.

### 3.1 Flow Matching

We base our generative models on the Flow Matching framework [26, 27], which defines a generation of data $\mathbf{X}_1 \sim p_d$ as the progressive transformation of $\mathbf{X}_0$ following a path connecting samples from the two distributions. Often, the path is instantiated as a linear interpolation: $\mathbf{X}_t = t\mathbf{X}_1 + (1-t)\mathbf{X}_0$, and $\mathbf{X}_0 \sim p_n = \mathcal{N}(0, \mathbb{I})$ originates from a noise distribution [27]. The velocity along this path is given by $v_t = \frac{d\mathbf{X}_t}{dt} = \mathbf{X}_1 - \mathbf{X}_0$. A generator $\mathcal{G}$ is trained to predict this velocity by minimizing:

$$\mathcal{L} = \mathbb{E}_{t \sim p_t, \mathbf{X}_1 \sim p_d, \mathbf{X}_0 \sim p_n} \left\| \mathcal{G}(\mathbf{X}_t, t) - v_t \right\|_2^2, \tag{1}$$

where $p_t$ is a training-time distribution over $t$. At inference, samples are generated from noise using an ODE solver (e.g., Euler), integrating the learned velocity field to produce data samples.

### 3.2 Input Representations for Progressive Modeling

The high dimensionality of visual modalities makes their modeling challenging. This complexity can be reduced by factorizing their generation into simpler *stages*, instantiating a progressive generation framework which generates coarse details such as structure first, and finer details later.

To this end, each input $\mathbf{X}$ is structured into a multiscale representation $\mathbf{P} = \{\mathbf{X}^s\}_{s=1}^S$ across $S$ scales, each defining a generative *stage*, containing a progressively increasing level of detail, from the most coarse to the finest. Various decomposition techniques are viable, including DWT, DCT, Fourier transforms, or multiscale autoencoders [8]. We adopt the Laplacian pyramid due to its simplicity and flexibility, but note that, unlike existing methods, our framework is agnostic to the particular choice of decomposition. The pyramid levels are defined as:

$$\begin{cases} \mathbf{X}^s = \mathrm{down}(\mathbf{X}, s) - \mathrm{up}(\mathrm{down}(\mathbf{X}, s-1), s) & \text{if } s > 1 \\ \mathbf{X}^1 = \mathrm{down}(\mathbf{X}, 1) & \text{if } s = 1 \end{cases}, \quad (2)$$

where each scale $s$ is associated with a user-defined resolution, and $\mathrm{down}(\dots, s)$ and $\mathrm{up}(\dots, s)$ respectively represent downsampling and upsampling to the resolution of $s$. A downsampled approximation of the input can be obtained as $\bar{\mathbf{X}}^{\mathbf{s}} = \mathbf{X}^1 + \dots + \mathbf{X}^s$.

### 3.3  Decomposable Flow Matching

The multiscale input is amenable to progressive modeling. We define a number of stages where each stage models a set of scales from the coarsest to the finest. We extend Flow Matching to support multistage generation. To this end, we introduce an independent flow timesteps $t^s$ for each scale, and formulate the forward process as:

$$\mathbf{P}_{t^1, \dots, t^S} = \{\mathbf{X}_{t^s}^s\}_{s=1}^S \quad \text{where} \quad \mathbf{X}_{t^s}^s = t^s \mathbf{X}_1^s + (1 - t^s)\mathbf{X}_0^s. \quad (3)$$

The factorization of flow timesteps enables control over stage generation by imposing conditional generation on previous stages and marginalization on successive stages by respectively applying low or full noise levels [2]. To this end, we define a discrete distribution $p_q$ over the number of sampling stages, and for each scale $s$ within the sampled stage $Q$, we draw timesteps independently from a predefined logit-normal distribution $p_{t,Q}^s$. Consequently, rather than predicting a single velocity, the model predicts per-scale velocities $v^s$ using a shared generator $\mathcal{G}$ conditioned on the full set of noisy input scales and their timesteps, which is learned by minimizing:

$$\mathcal{L} = \mathbb{E}_{Q \sim p_q, \, t^1 \sim p_{t,Q}^1, \dots, t^S \sim p_{t,Q}^S, \, X_1 \sim p_d, \, X_0 \sim p_n} \sum_{s=1}^{S} M^s \left\| \mathcal{G}_s\left(P_{t^1, \dots, t^S}, t^1, \dots, t^S\right) - v^s \right\|_2^2. \quad (4)$$

where $p_q$ represents the a discrete distribution over the number of stages, $Q$ represents the sampled stage, $p_t$ represents the training distribution of each scale timesteps, and $\mathcal{G}_s$ represents the the model velocity prediction for scale $s$.

During inference, a timestep schedule is defined that activates stages progressively as shown in Figure 2a. The sampler starts from the first stage and advances $t^1$ only, before introducing subsequent scales one at a time once the last introduced scale reaches a predefined threshold $\tau$ in its generation. Then, the last stage along with all previous stages, continues the generation at different rates (see Sampling Schedule in Figure 2 (a)). Each time the generated scale reaches its threshold, the representation can undergo partial decoding up to the current stage, producing an output preview. At the end of the generation, all scales representations are summed to visualize the generated output.

Both the stage distribution $p_q$ and the training timestep distribution $p_t$ are central to effective learning of this progressive strategy, and training timesteps are chosen to simulate the progressive sampling through stages, as shown in Figure 2b. First, a stage index, representing the number of stages that are currently under generation, is sampled from a discrete probability distribution $p_q$. Given the stage number, we sample the timestep of the last scale within the stage according to a logit normal distribution [7]. For every preceding stage, we instead sample the respective timestep from a logit normal distribution shifted towards lower noise levels to simulate inference time behavior where the earlier stages are less noisy than the last stage currently being generated. For simplicity, we apply the same shift to all preceding scales, using the same logit-normal parameters across them. Finally, the timesteps for each subsequent stage are set to zero to prevent the model from spending its capacity in modeling such zeroed scales. Algorithm 1 shows the procedure for sampling the stage index and the timesteps for each scale within the stage. We additionally introduce a masking term $\mathbf{M}^s$ that ignores their loss contributions. This promotes progressive learning dynamics, where the model focuses on global structure before local detail, enabling high-quality progressive generation.

### 3.4  Architectural Adaptations

We choose to experiment on a DiT architecture [33] due to the pervasive adoption of its variants [7, 23, 25, 28, 46, 48], and we extend it to accommodate multiscale inputs and their associated flow

Table 1: Ablation results on Imagenet-1K [5] using a DiT-B backbone.

| | Res. | $\text{FID}_{10K}\downarrow$ | $\text{FDD}_{10K}\downarrow$ | IS $\uparrow$ |
|---|---|---|---|---|
| *(a) First stage samp. prob. $p_t^0$* | | | | |
| 0.7 | 512 | 36.02 | 676.0 | 32.2 |
| 0.8 | 512 | 32.09 | 622.0 | 36.1 |
| 0.9 | 512 | 29.56 | 571.0 | 41.0 |
| 0.95 | 512 | 27.5 | 540.0 | 43.7 |
| *(b) Previous scale lognorm location* | | | | |
| 0.0 | 512 | 29.83 | 567.0 | 40.4 |
| 0.5 | 512 | 28.89 | 569.0 | 41.5 |
| 1.0 | 512 | 29.14 | 567.0 | 41.2 |
| 1.5 | 512 | 29.56 | 571.0 | 41.0 |
| 2.0 | 512 | 29.96 | 589.0 | 39.8 |
| Tied | 512 | 55.49 | 885.0 | 21.4 |

| | Res. | $\text{FID}_{10K}\downarrow$ | $\text{FDD}_{10K}\downarrow$ | IS $\uparrow$ |
|---|---|---|---|---|
| *(c) Architecture* | | | | |
| baseline | 512 | 29.56 | 571.0 | 41.0 |
| −masking | 512 | 29.9 | 584.0 | 40.0 |
| −standardization | 512 | 27.13 | 505.0 | 42.8 |
| *(d) Parameter Specialization* | | | | |
| None | 512 | 28.37 | 564.0 | 41.5 |
| Mod. | 512 | 28.95 | 568.0 | 41.4 |
| Proj. | 512 | 29.56 | 571.0 | 41.0 |
| Cond. | 512 | 28.88 | 565.0 | 40.0 |
| Attn. | 512 | 28.99 | 564.0 | 40.7 |
| MLP | 512 | 27.65 | 545.0 | 43.0 |
| Full | 512 | 26.63 | 526.0 | 45.5 |

| | Res. | $\text{FID}_{10K}\downarrow$ | $\text{FDD}_{10K}\downarrow$ | IS $\uparrow$ |
|---|---|---|---|---|
| *(e) Compute Allocation* | | | | |
| ↑ Batch size | 512 | 31.95 | 615.0 | 39.9 |
| ↓ Patch size | 512 | 29.56 | 571.0 | 41.0 |
| *(f) Input Decomposition* | | | | |
| 512→1024 | 1024 | 58.03 | 899.0 | 20.3 |
| 256→512→1024 | 1024 | 40.60 | 683.2 | 31.8 |
| 256→1024 | 1024 | 41.06 | 704.0 | 30.9 |

timesteps (see Figure 2c). At their core, a DiT transforms the input visual modality into a sequence of tokens through a linear layer called *patchification* layer, where each token corresponds to a small portion of the input of size $k \times k$, denoted as *patch size*. An inverse operation is performed by the output patchification layer. A sequence of transformer blocks [45] processes the token sequence. To make the model aware of the input timestep, a *timestep embedder* MLP produces a respective conditioning embedding, which is passed to the transformer blocks through modulation layers.

To accommodate multiscale inputs, rather than using single input and output patchification layers, dedicated input and output patchification layers are instantiated per scale. Patch sizes are chosen such that each input scale yields an equal number of patches, ensuring consistent spatial alignment across scales. For example, for two stages of resolution $256 \times 256$ and $512 \times 512$, we use patch sizes of $1 \times 1$ and $2 \times 2$, respectively. The resulting patch embeddings from all scales are summed and fed into the transformer backbone. Finally, a dedicated per-scale projection layer is used to output each scale velocity. Loss masks $\mathbf{M}$ are integrated to zero out model inputs at masked scales. This prevents noisy inputs whose contribution to the loss is negated by the mask from harming model performance. To incorporate the per-scale timesteps, we use a dedicated timestep embedder for each level and sum their outputs to form a unified conditioning signal.

## 4 Experiments

### 4.1 Experimental setup

**Latent space** The FLUX [23] and Cogvideo [48] autoencoders are used, respectively, to produce the latent spaces for all image and video experiments. To maximize the benefits of coarse-to-fine modeling, low-, mid-, and high-frequency components in the latents should correspond to the respective low-, mid-, and high-frequency RGB components once decoded, a property not typically found in modern autoencoders. We establish it for all autoencoders by scale-equivariant finetuning [38]. Namely, we fine-tune the FLUX autoencoder for 10,000 steps on our internal image dataset at $256^2$-resolution with $\times$2-4 scale equivariance regularization [38] with a batch size of 64.

**Training details** We train our models on the ImageNet-1K [5] and Kinetics-700 [3] datasets. We run all ablations using DiT-B and main experiments on the DiT-XL architecture. All experiments use a batch size of 256, a learning rate of $1e{-}4$ with 10k steps warmup, gradient clipping of 1.0, the Adam [21] optimizer, and an EMA with $\beta = 0.9999$. We train ablations on ImageNet-1K for 600k steps, and the main experiment for 500k and 350k steps, respectively, for 512px and 1024px resolution. For Kinetics-700, we train the main experiments for 200k steps.

**Evaluation metrics and protocol** We report FID, and Inception Score (IS) for image experiments and frame-FID and FVD [43] for video experiments. We also include FDD (Frechet Distance computed against DINOv2 [32] features) as it was shown to be a more reliable metric than FID [20, 39]. We use 50k generated samples for ImageNet-1K 512px evaluation and 10k samples for the rest of the experiments. We report main experiment results with cfg [12] values of 1.0, 1.25, and 1.5 as [33].

### 4.2 Training Parameters Ablation

We conduct ablations to provide insights into the behavior of the main training hyperparameter. Unless otherwise specified, the ablations are run on ImageNet-1K [5] 512px using a DiT-B backbone. The default backbone employs a 2-stage Laplacian decomposition of 256px and 512px resolution,

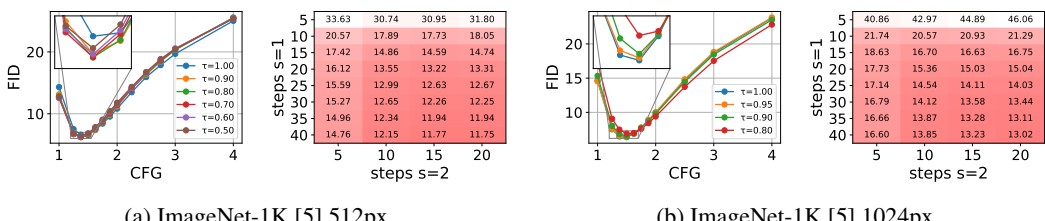

(a) ImageNet-1K [5] 512px   (b) ImageNet-1K [5] 1024px

Figure 3: $\text{FID}_{10K}$ results ablating sampling configuration, threshold, and per-stage steps.

Table 2: DFM efficiency comparison with Flow Matching

| Model | # Params | Peak Memory | GFLOPs | Forward Latency | Train Iter Speed | Sampling Time |
|---|---|---|---|---|---|---|
| Flow Matching | 712.122 M | 6.258 GB | 524.613 | 60.25 ms | 297.39 ms | 2.43 s |
| DFM | 716.482 M | 6.294 GB | 524.638 | 61.32 ms | 301.00 ms | 2.43 s |

respectively, where each decomposed input scale is normalized to unit standard deviation. Results are presented in Table 1, and practical guidance on their optimal configuration is provided in *Appendix* Sec. A.5.

**Training timestep distribution.** The sampling distribution of timesteps $p_t$ during training is crucial as it determines the implicit loss weighting for each stage (see Section 3.3). We parametrize its design across two dimensions. The first is the distribution over the stage index $s$, which corresponds to a number of scales being trained. Since we ablate over a 2-stage configuration, this can be expressed as the probability of sampling stage 0 ($p_q^0$). The second is the location for the logit normal distribution [7] of all preceding stages, representing the amount of bias towards lower noise levels for already-generated stages. We report the performance of various sampling strategies in Table 1 (a) and (b). Performance is positively impacted by large probabilities of sampling stage 0, indicating that allocating more model capacity towards predicting structural details benefits generation. This is aligned with previous cascaded model training strategies, where the first stage model is trained longer than the second stage model [29]. Performance is also positively impacted by larger shifts in the logit normal distribution of already-generated scales through previous stages, as this better simulates the inference-time sampling schedule. We found that location values of 0.5, 1.0, and 1.5 yield comparable performance, with differences of less than 1% in FDD and 1.3% in IS. A larger location value, however, simulates availability of a cleaner first scale (low-frequency) result and aligns more closely to inference time behavior. We hypothesize that under the increased DiT/XL capacity used in the main experiments, a closer alignment to inference-time behavior would be beneficial as we speculate that larger models may afford spending more capacity for non-structural details. Therefore, we select $p_q^0 = 0.9$ and location of 1.5 for our main experiments. Finally, we consider a strategy that ties the timesteps of every stage, enforcing the same value for them, and accordingly modify sampling. This variation presents degraded performance as it loses the benefits of progressive generation.

**Architecture.** We ablate our baseline by removing input masking (see Section 3.4) and removing the normalization of each stage to unit standard deviation before the application of the diffusion process. We report the results in Table 1 (c). Removing input masking for successive stages degrades performance over all metrics due to the negative influence of injecting pure noise tensors into the model. However, we found that removing standardization improves performance. Removing normalization results in a lower magnitude for the input of the second stage. This affects the model in a similar way to increasing $p_q^0$, giving more importance to stage 1 structural modeling and improving performance. Therefore, we adopt masking but exclude standardization for our main experiment.

**Parameter Specialization.** We leverage the factorization of generation into multiple stages by exploring model parameter specialization for each stage. Specifically, in addition to base model parameters, we create a set of per-stage model parameters acting as *stage experts*, akin to using distinct sets of model parameters in cascaded models. During the generation of stage $s$, the specialized version of the weights corresponding to $s$ is averaged with the base model parameter, enabling scale-specific behavior without significant computational overhead. As shown in Table 1 (d), parameter specialization improves performance when applied to MLP layers and produces the best results when applied to every model parameter. Despite its performance benefit, we exclude this technique from our main experiments to keep the parameter count similar across baselines.

Table 3: Comparison to baselines on ImageNet-1K [5] and Kinetics-700 [3] using a DiT-XL model and training convergence plot for ImageNet-1K [5] 512px.

| | *ImageNet 512px* | | | *ImageNet 1024px* | | | *Kinetics 512px* | | |
| | $FID_{50K}\downarrow$ | $FDD_{50K}\downarrow$ | IS $\uparrow$ | $FID_{10K}\downarrow$ | $FDD_{10K}\downarrow$ | IS $\uparrow$ | $FID_{10K}\downarrow$ | $FDD_{10K}\downarrow$ | $FVD_{10K}\downarrow$ |
|---|---|---|---|---|---|---|---|---|---|
| Flow Matching | 15.89 | 282.9 | 58.5 | 42.18 | 575.9 | 26.7 | 12.6 | 370.1 | 304.1 |
| Cascaded | 13.67 | 260.9 | 72.7 | 20.16 | 318.3 | 55.0 | 13.81 | 346.2 | 317.2 |
| Pyramidal Flow | 10.98 | 261.2 | 77.3 | 16.9 | 351.1 | 64.0 | 10.51 | 353.8 | 265.6 |
| DFM (Ours) | 9.77 | 200.6 | 77.9 | 14.11 | 277.3 | 78.2 | 9.98 | 336.7 | 260.2 |
| Flow Matching (cfg=1.25) | 7.59 | 185.7 | 97.2 | 31.57 | 405.3 | 39.4 | 10.1 | 312.9 | 297.0 |
| Cascaded (cfg=1.25) | 6.80 | 163.0 | 128.4 | 11.68 | 221.9 | 89.2 | 11.62 | 279.9 | 308.8 |
| Pyramidal Flow (cfg=1.25) | 4.95 | 164.7 | 132.0 | 9.38 | 245.7 | 109.3 | 8.24 | 292.2 | 266.3 |
| DFM (Ours) (cfg=1.25) | 4.28 | 120.0 | 135.6 | 7.56 | 181.7 | 130.2 | 7.88 | 277.5 | 257.4 |
| Flow Matching (cfg=1.5) | 4.73 | 132.5 | 138.9 | 24.68 | 412.5 | 46.0 | 8.43 | 268.4 | 309.8 |
| Cascaded (cfg=1.5) | 5.68 | 114.0 | 118.1 | 8.23 | 167.3 | 126.8 | 10.39 | 235.0 | 327.1 |
| Pyramidal Flow (cfg=1.5) | 4.57 | 116.2 | 191.0 | 7.68 | 185.9 | 152.3 | 6.98 | 247.1 | 280.2 |
| DFM (Ours) (cfg=1.5) | 4.28 | 85.8 | 196.8 | 6.42 | 132.5 | 186.5 | 6.85 | 236.5 | 271.7 |

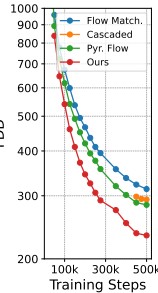

**Compute allocation.** Compared with a single-stage model, input decomposition entails a lower dimensionality for earlier stages which results in saved computation at similar settings. In the ablations, for example, the first stage corresponds to 256px inputs rather than 512px. Such computational savings can be leveraged by either increasing the training batch size and number of inference steps or reducing the DiT patch size so that each stage results in an equal number of tokens. We observe performance increases at a steady rate when reducing patch size, yet increasing the batch size quickly reaches a plateau, converging to lower performance as reported in Table 1 (e). We select the patch size reduction strategy for the final model.

**Input Decomposition.** Our framework is agnostic to the exact choice of input decomposition, and thus different number of stages or different resolutions at different stages can be seemingly applied. We explore different configurations of the Laplacian decomposition for 1024px ImageNet-1k generations and report the results in Table 1 (f). Each decomposition is denoted as the sequence of resolutions corresponding to each stage *e.g.* 256→1024 for a 2-stage decomposition with 4x downsampling between stages. The 512→1024 decomposition yields the worst performance due to the initial stage being performed in 512px resolution and containing a larger amount of non-structural components, reducing the effectiveness of the progressive framework. We find the three-stage setting (256→512→1024) to yield the best result. However, we adopt 2-stage decompositions of 256→512 and 256→1024, respectively, for 512px and 1024px experiments for easier comparison with baselines.

## 4.3 Sampling Parameters Ablation

This section analyzes the effects of sampling parameters on a DiT-XL trained on ImageNet-1k 512px and 1024px using the optimal training parameters discussed in Section 4.2. Results are given in Figure 3, and practical guidance on their optimal configuration is provided in the *Appendix* Sec. A.5.

**Stage Timestep Thresholds.** We analyze the optimal timestep threshold $\tau$ determining the switch between sampling stages. We choose a threshold of 0.7 for 512px as it produces the best FDD and second-best FID at the respective optimal cfg level and 0.95 for 1024px experiments.

**Per-Stage Sampling Steps.** We evaluate model performance using different numbers of steps allocated to the sampling of the first and second stages under a sampling threshold of 0.7 and 0.95 respectively for 512px and 1024px. Allocating a larger number of steps to the first stage gives overall better performance, matching the intuition that the generation of structural details represents a more crucial component of sample quality than fine-grained details. We fix a total number of steps equal to 40 for our experiments and choose 30 and 10 sampling steps, respectively, for the two stages.

## 4.4 Comparison to baselines

**Baselines selection and details.** We select baselines corresponding to the main generation paradigms in Figure 1, which we train on the same underlying model backbone, ensuring a constant amount of training compute. The cascaded model baseline consists of separate models, one for each stage. Two stages are employed to avoid excessive error accumulation. We train the first stage model for 80% of the total training iterations and use it to initialize the second stage model for faster convergence. Pyramidal Flow [18] is chosen due to its state-of-the-art progressive generation performance and due to it operating in latent spaces while using a rectified flow formulation. We operate it in a 2 and 3-stage configuration for 512px and 1024px datasets, ensuring the same base resolution across

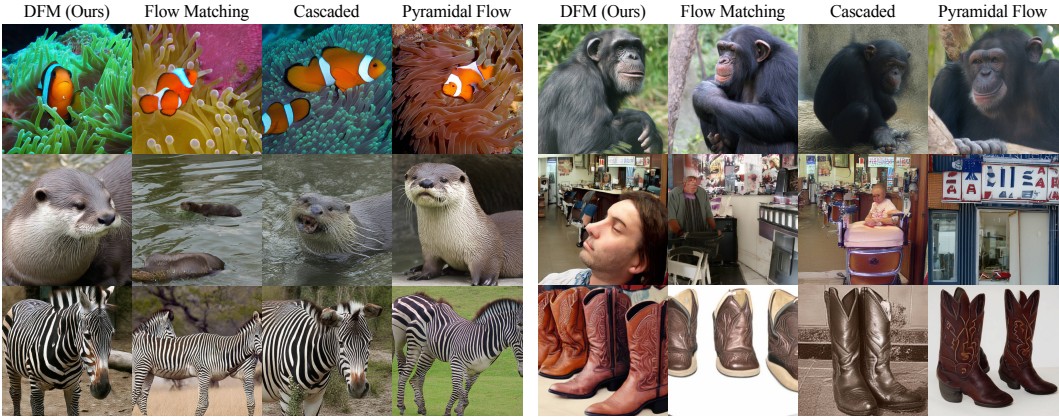

Figure 4: Comparison of DFM against baselines on ImageNet-1K 512px [5] on selected samples. Samples are generated with cfg 3.0

experiments. All baselines are configured to ensure the same amount of training compute. Please refer to *Appendix* Sec. B for exact details on each baseline.

**Results.** As shown in Table 3, our method surpasses the baselines across all metrics and cfg settings with the exception of Kinetics-700 [3] FDD. Notably, DFM matches baselines best reported FDD in roughly half of the number of iterations, while using the same training compute (See Figure 4, right) Qualitative comparison (see Figure 4 and *Appendix* Sec. C) suggests that DFM generates better structural details, an observation which we attribute to its progressive formulation which focuses the model's capacity on its lowest resolution stage.

**Efficiency.** To ensure a fair comparison, we keep the training compute budget comparable across all baselines in Table 3. We further benchmark DFM against Flow Matching in Table 2 on DiT-XL. As shown, the additional patchification and timestep-embedding layers introduced by DFM incur only a minor increase in parameters and training FLOPs, translating to less than 1.2% slowdown per training iteration.

### 4.5 Finetuning of large-scale models

To evaluate the effectiveness of DFM in large-scale generative models, we finetune FLUX-dev [23] on our internal image dataset using DFM and scale equivariant autoencoder. First, the original FLUX model is finetuned for 24k steps to adapt it to the scale equivariant autoencoder. Starting from this checkpoint, we branch two finetunings. The first continues regular finetuning to obtain the FLUX-FT checkpoint, the second finetunes for our framework to obtain the FLUX-DFM

Table 4: FLUX fine-tuning comparison.

| Method | $FID_{10K} \downarrow$ | $FDD_{10K} \downarrow$ | CLIPSIM $\uparrow$ |
|---|---|---|---|
| FLUX | 17.79 | 172.7 | 0.3280 |
| FLUX-FT | 12.56 | 153.6 | 0.3261 |
| FLUX-DFM | 8.83 | 116.2 | 0.3381 |

checkpoint. Both are finetuned for 32k steps under the same settings and training compute. We use FLUX-dev patchification and projection layers weights to initialize the first stage layers of FLUX-DFM, and zero-initalize the second stage layers. We train on multiple aspect ratios and use a total batchsize of 192 for the 1024px resolution. More details on the finetuning are included in the *Appendix* Sec. E. We compare the baselines by generating 10k samples from a validation split of our internal dataset and report FID, FDD, and CLIP [34] similarity (CLIPSIM) in Table 4. We include qualitative results and comparisons for FLUX-DFM in Figure 5 and *Appendix* Sec. C. FLUX-DFM achieves consistently better performance compared with standard full finetuning, improving FID and FDD by 28.7% and 24.3%, respectively. We report original FLUX-dev checkpoint numbers as a reference. Please note that in this experiment, we do not aim to improve FLUX-dev quality but rather show that finetuning with DFM achieves overall faster convergence in learning the finetuning dataset distribution. Thus, report FID and FDD as reliable metrics comparing FLUX-FT and FLUX-DFM.

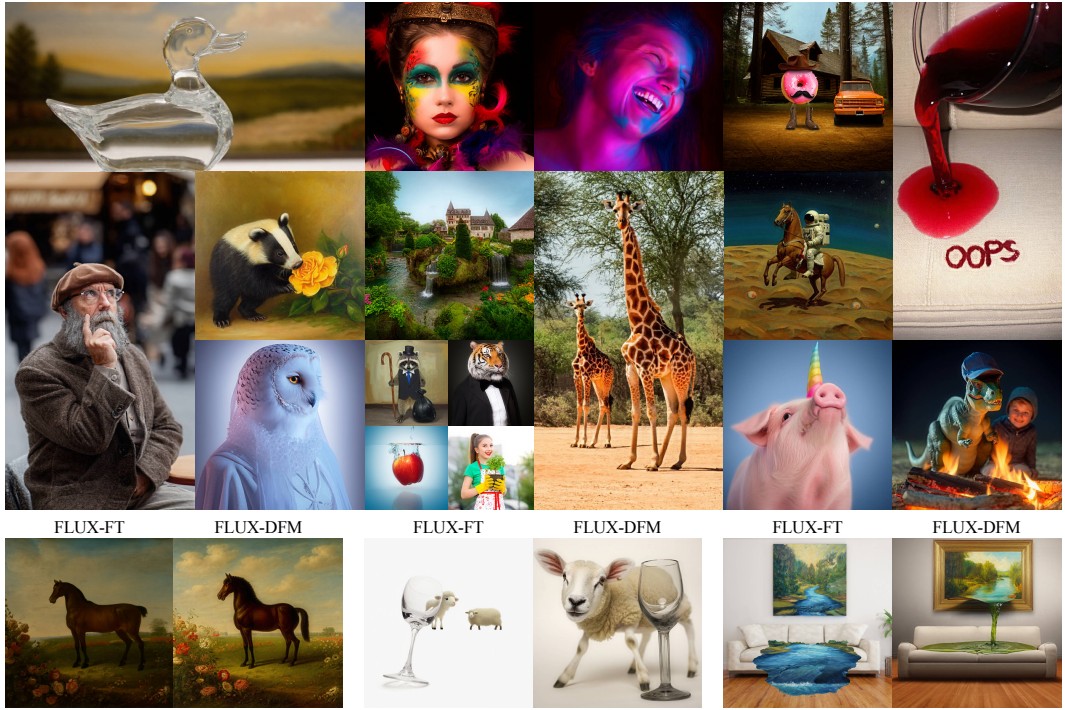

Figure 5: (Top) Qualitative results produced by FLUX [23] finetuned with DFM. (Bottom) a comparison of FLUX-DFM to the baseline (FLUX-FT).

## 5 Discussion

**Limitations** Similarly to other progressive generation frameworks [1, 9, 18, 41] our method introduces additional training and sampling hyperparameters. The framework's most important hyperparameters balance between learning of low and high-frequency visual components, enabling users to pose more emphasis on structural details. When excessive importance is put on structural details, a decreased presence of high-frequency details is noticed. We provide extensive ablations, and practical guidance for their setup, and show that the same parameters can be reused across different settings. We provide further ablations and guidance in *Appendix* Sec. A and discuss further our framework limitations and failed experiments in *Appendix* Sec. F and *Appendix* Sec. G

**Conclusions** We propose DFM, a novel framework for the progressive generation of images and videos. Differently from existing formulations, our framework is simple, agnostic to the choice of decompositions and does not necessitate multiple networks. We analyze it to provide insights into its main training and sampling hyperparameters and show that it outperforms both Flow Matching and the state-of-the-art progressive diffusion formulations on both ImageNet-1K [5] and Kinetics-700 [3]. We further validate the effectiveness of DFM on large-scale models, showing its faster convergence than standard full fine-tuning. A promising avenue for future work is to explore other decomposition paradigms such as discrete cosine, wavelet transforms, and multiscale autoencoders. Extending these findings to large-scale settings remains an exciting direction for future research.

**Acknowledgments** The authors thank Anil Kag and Dishani Lahiri for their engineering support.

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

# Appendix

## Appendix Contents

## A   Framework Details

### A.1   DFM Implementation Details

We base our architecture on a modified version of DiT [33]. Specifically, we normalize the data across scales by applying scale-wise pre- and postconditioning following the scheme of [19], which ensures that each input and output distribution has unit variance in expectation. Additionally, we replace ViT frequency-based positional embeddings applied in DiT [33] with 2D rotary positional embeddings (RoPE) [40] due to its wide adaptation in recent models [23, 46]. Furthermore, we condition the model on the stage being generated by adding an embedding of the current stage index to the modulation signal. Finally, we drop the class label conditioning $10\%$ of the time to enable classifier-free guidance.

To adapt the DiT [33] for video generation, we patchify the latent frames independently and apply tokenwise concatenation of the resulting tokens [46]. Furthermore, we replace the 2D RoPE with 3D RoPE to adapt the position information to the video input. Finally, our ablations on Kinetics-700 reveal that unlike the FLUX autoencoder (See Table. 1 (c)), Cogvideo latents benefit from standardizing the input to have unit variance before the application of the diffusion process. Therefore, we apply such standardization to all of our video experiments.

### A.2   DFM Training Details

We train using the Adam optimizer ($\beta_1 = 0.9$, $\beta_2 = 0.99$, $\epsilon = 10^{-8}$) and a base learning rate of 0.0001 with 10k linear warmup steps, weight decay of 0.01, and total batchsize of 256. The main experiments are trained with DiT-XL/2 on ImageNet-1K for 512px and 1024px, respectively, for 500k and 350k steps. For the main video experiments on Kinetics-700, we train for 200k steps. Ablations are trained with DiT-B/2 for 600k steps using the same hyperparameters as the main experiments.

ImageNet-1K 512px experiments are trained on a single node containing 8 H100 GPUs, whereas ImageNet-1K 1024px and Kinetics-700 512px experiments used 2 nodes of the same type. Ablations were trained on a single node.

### A.3 DFM Dataset Details

Our main image experiments are trained and evaluated on ImageNet-1K [5] which has a research-only, non-commercial license. Our video experiments are trained and evaluated on Kinetics-700 [3], which is available under the Creative Commons Attribution 4.0 (CC BY 4.0) license.

### A.4 Inference Details

For all of our experiments, unless otherwise specified, we use an Euler ODE sampler with 40 steps and a linear timestep scheduler.

### A.5 Choosing Training and Inference Hyperparameter

DFM introduces several training and inference hyperparameters. In practice, a default configuration of such hyperparameters generalizes well across different datasets and settings. In the following, we discuss such optimal configurations and summarize the observed effects of varying the main hyperparameters.

- **Number of generation stages**: Using 2 stages works well for most experiments, where the first stage has a resolution of 256px, while the second stage has a resolution equal to the final resolution (see Table. 1). We find that a base resolution of 256px for the first stage contains an ideal amount of structural detail to support generation of the successive ones while not containing excessive amounts of fine-grained details.

- **First stage sampling probability** $p_t^0$: We find that 0.9 generalizes well across different model sizes. The probability determines the amount of model capacity spent on structural detail modeling, so smaller models may benefit from higher values (see Table. 1)

- **Stage noise sampling parameters**: At training time, we use a logit normal distribution with parameter ($location$=0, $scale$=1.0) for the stage currently sampled for training, and ($location$=1.5, $scale$=1.0) for the previous stage. Larger location values for previous stages allow the model to leverage more structural details as conditioning during second stage generation, but expose the model more to train-inference mismatches if the first stage results are not generated with sufficient quality. Thus, larger location values are more beneficial for larger models (see Table. 1).

- **Sampling stage threshold** $\tau$: 0.9 generalizes well across different settings (see Figure 6). As the parameter determines the amount of information in the first stage that the second stage can leverage as conditioning, in the presence of a high-quality first stage prediction, larger thresholds are desirable. While the optimal value varies depending on cfg, model, and dataset, we find the suggested value to be a reliable default.

- **Input standardization:** Before applying noise, we standardize each level in the decomposed input representation to have unit-variance for Kinetics-700 video experiments using the CogVideo autoencoder, while this behavior is disabled for ImageNet-1K image generation experiments with the FLUX autoencoder. Standardization has an impact on loss magnitude for each stage, thus it acts similarly to the first stage sampling probability $p_t^0$ in balancing the amount of model capacity allocated to modeling structural details or fine details. We found that the optimal parameters can vary depending on the autoencoder representation and we suggest ablating over this setting when adopting a new autoencoder.

## B   Baseline Details.

This section, includes details about the baselines training and inference. First, the base architecture is described, then specific details about the Cascaded and Pyramidal Flow baselines are detailed.

Table 5: DFM with original (orig) and scale-equivariant (SE) FLUX autoencoder. Experiments are performed on DiT-B and trained on ImageNet-1K [5] 512px for 400k steps.

| Method | $FID_{50K} \downarrow$ | $FDD_{50K} \downarrow$ | $IS \uparrow$ |
|---|---|---|---|
| Flow Matching (FLUX-ae-orig) | 54.50 | 855.5 | 21.4 |
| DFM (FLUX-ae-orig) | 34.00 | 630.1 | 34.2 |
| Flow Matching (FLUX-ae-SE) | 43.16 | 728.85 | 26.6 |
| DFM (FLUX-ae-SE) | 32.89 | 626.5 | 35.0 |

Table 6: Comparison of various decomposition methods on ImageNet-1K [5] 512px. Resutls at different with different guidance (CFG) values are reported.

| | CFG 1.0 | | | CFG 1.25 | | | CFG 1.5 | | |
|---|---|---|---|---|---|---|---|---|---|
| | $FID_{50K} \downarrow$ | $FDD_{50K} \downarrow$ | $IS \uparrow$ | $FID_{50K} \downarrow$ | $FDD_{50K} \downarrow$ | $IS \uparrow$ | $FID_{50K} \downarrow$ | $FDD_{50K} \downarrow$ | $IS \uparrow$ |
| Flow Matching | 15.89 | 282.9 | 58.5 | 7.59 | 185.7 | 97.2 | 4.73 | 132.5 | 138.9 |
| Ours (DWT) | 9.48 | 195.6 | 78.8 | 4.20 | 117.3 | 136.9 | 4.40 | 84.8 | 199.2 |
| Ours (DCT) | 9.60 | 194.4 | 78.6 | 4.19 | 115.1 | 138.3 | 4.37 | 82.5 | 201.2 |
| Ours (Laplacian) | 9.77 | 200.6 | 77.9 | 4.28 | 120.0 | 135.6 | 4.28 | 85.8 | 196.8 |

**Baselines architecture.** All baselines use the hyperparameters and architecture of [33], with the modification to incorporate rotary positional embeddings (RoPE) [40] and Network Precondtioning [19]. Inference is performed with 40 steps using an Euler sampler and a linear sampling schedule.

**Cascaded baseline.** We initially train the stage 1 model following the spatial baseline for $\approx 80\%$ of the total training steps. To match the training and inference compute of the other baselines, we adopt a patch size of $1 \times 1$, yielding the same number of tokens as the other baselines. Subsequently, we finetune the stage 1 model to obtain the stage 2 model, which upsamples the first stage output. For the stage 2 model, we introduce a dedicated pacification layer for the conditioning input from stage 1 and concatenate its output tokens with those in stage 2. We add a small amount of noise to the conditioning input during training to reduce exposure bias. The amount of noise is sampled from a logit normal distribution with scale parameters of 1.0 and location of 1.5. For inference, we perform a grid search over the best inference parameters and found that equally dividing the inference steps between stage 1 and stage 2 (*i. e.* 20 inference steps for each stage) and using a noise level of 0.025 yields the best results. Therefore, we use these settings for our evaluations.

**Pyramidal Flow baseline.** We follow Jin *et al* [18] in training the Pyramidal Flow baseline. For the three-stages experiments, we allocate twice the batch size for the second stage compared with the first and second stage following [18] and use a gamma parameter of 0.33. In the two-stages experiments, we allocate an equal batch size between the first and second stages. During inference, we use an equal number of inference stapes for each stage.

## C    Additional Evaluation Results and Details

**Choice of decomposition method.** DFM employs a multi-scale spatial representation that explicitly separates low- and high-frequency content. This separation can be realized with several decomposition methods(e.g., Laplacian pyramid, discrete cosine transform (DCT), or discrete wavelet transform (DWT)) by decoding low- and high-frequency components independently. Despite their representational differences, we find the practical guidance on choosing the hyper-parameters in section A.5 still applies. We retrained DFM on ImageNet-1K at 512px with DCT- and DWT-based decompositions using exactly the settings in Table 3, and report results in **??**. As shown, using identical hyper-parameters yields comparable gains over the Flow-Matching baseline across all decompositions, indicating that DFM is agnostic to the choice of decomposition and that the default hyper-parameters are broadly transferable.

**Sampling parameters ablation** We report quantitative results ablating the role of different sampling hyperparameters in Figure 6 and show qualitative results in Figure 18 and Figure 19 ablating, respectively, sampling steps allocated to each stage, and different cfg and sampling threshold configurations. Increasing the number of steps allocated to the first stage results in improved image structure, while

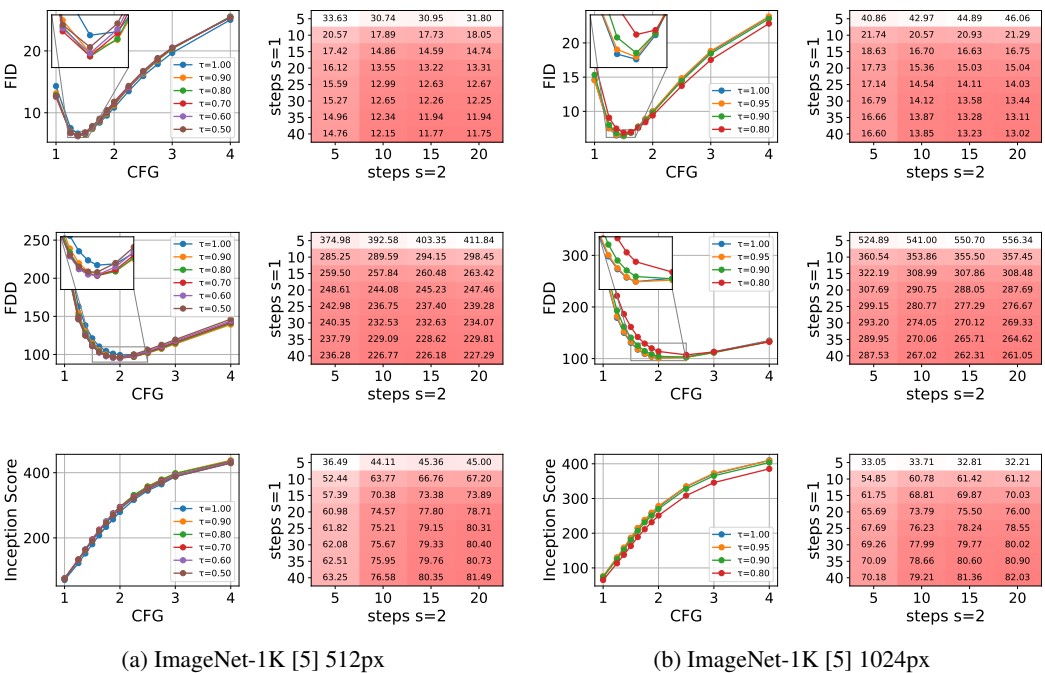

(a) ImageNet-1K [5] 512px        (b) ImageNet-1K [5] 1024px

Figure 6: $FID_{10K}$, $FDD_{10K}$ and IS ablating sampling configuration, threshold, and per-stage steps on DiT-XL/2 trained with DFM on ImageNet-1K [5].

increasing the number of steps allocated to the second stage results in reduced gains which are observable in areas with complex textures such as foliage, grass, or animal fur. The sampling threshold has a reduced impact on generation quality and is most visible at low cfg values.

**Comparison to baselines** Convergence behavior for DFM and baselines is shown in Figure 8, Figure 9, and Figure 10, respectively, for ImageNet-1K [5] 512px, ImageNet-1K [5] 1024px, and Kinetics-700 [3]. In addition, we show qualitative comparison on non-curated samples for DFM against baselines on ImageNet-1K [5] 512px (see Figure 11), ImageNet-1K [5] 1024px (see Figure 12), and Kinetics-700 [3] (see Figure 13, Figure 14, and Figure 15). Addtional qualitative results and videos are provided in the *Website*.

**FLUX-DFM qualitative comparison with FLUX-FT**. We provide additional qualitative comparison of FLUX finetuned with DFM (FLUX-DFM) against FLUX with standard finetuning applied (FLUX-FT) on 1024px text-to-image generation in Figure 16 and Figure 17.

**Qualitative results.** We provide qualitative results on ImageNet-1K [5] 512px of selected classes in Figure 20, Figure 21, and Figure 22. Addtionally, we provide fully uncurated samples in Figure 23, Figure 24, and Figure 25.

# D   Autoencoders without Scale Equivariance

DFM explicitly decouples low- and high-frequency content into two successive stages. The first stage diffuses coarse, low-frequency information. Its output then conditions the generation of high-frequency details in the second stage. Recent work shows that diffusion models follow this spectral autoregressive pattern implicitly [30, 36, 38] and proposed scale-equivariant autoencoders to improve this separation, making the produced latents easier to diffuse [38]. Although DFM is agnostic to the choice of autoencoder, we conduct our experiments with the autoencoders of [38] finetuned for scale equivariance to ensure desirable spectral properties both for baselines and DFM.

In this section, we explore the behavior of DFM on non-regularized autoencoders. Table 5 reports a comparison between Flow Matching and DFM on the original FLUX autoencoder and scale-equivariant FLUX autoencoder for a DiT-B architecture trained for 400k steps on ImageNet-1K [5]. When the scale-equivariant FLUX autoencoder is replaced with the original variant, Flow Matching

suffers a substantial drop in performance, whereas DFM preserves a similar performance, thanks to its explicit spectral decomposition. In both cases, DFM outperforms Flow Matching.

# E    Large Scale Finetuning

This section explores the usage of DFM for fine-tuning large-scale models. Since the stage 1 input resembles a low-resolution version of the original input, we aim to preserve the pretrained model's behavior as much as possible. DFM adds a per-stage patchification layer, timestep embedding, and output head. Accordingly, we reuse the pretrained patchification layer and timestep embedding for stage 1, while zero-initialising the corresponding stage 2 modules. Before training, the framework therefore replicates the pretrained model's low-resolution predictions. After training, all weights (including patchification layers) are adapted to generate consistent stage 1 and stage 2 outputs. To adjust the model patchfication layers weight to stage 1, which uses a smaller patch size, we upsample stage 1 input by a factor of 2 before adding noise, halving the effective patch size. We then downsample the model's output to match stage 1 resolution.

## E.1    FLUX DFM Finetuning

We choose to fine-tune FLUX-DEV [23] due to its strong performance. To avoid biases introduced by cfg distillation, the distillation-guidance factor is fixed at $1.0$ throughout training, and all fine-tuning is carried on an internal dataset. We follow [38] to regularize the the FLUX autoencoder, yielding a *scale-equivariant* autoencoder.

Then, we perform full-finetuning of FLUX for 24k steps to adjust it to the new autoencoder, producing FLUX-SE. During the initial 4k training steps, we freeze all layers except the patchfication layers. Then, we finetune FLUX-SE for 32k steps to obtain FLUX-DFM. As a baseline, we finetune FLUX-SE for the same 32k steps using standard full-finetuning. Finetuning uses 1024-px and 512-px images at variable aspect ratios. We use Adam optimizer ($\beta_1 = 0.9$, $\beta_2 = 0.99$, $\epsilon = 10^{-8}$) and a base learning rate of $0.00001$ with 2k linear warmup steps, weight decay of $0.01$, and total batch size of 192. We drop the text conditioning $10\%$ of the time to enable classifier-free guidance.

Please note that since FLUX-DEV is distilled and post-trained on highly aesthetic images, its distribution differs from that of our internal data. Therefore, a direct comparison with the original FLUX-DEV would not be informative. Rather, we measure speed in learning the new training distribution with DFM compared with standard full-finetuning at an equal training cost.

**Inference.** We evaluate using a test split of 10k prompts from our internal dataset. We use 40 sampling steps and Euler ODE solver, cfg of 3.0, and a distillation guidance factor of $1.0$ (equivalent to not applying cfg).

# F    Failed Experiments

**DCT-Space DFM** We apply DFM in the DCT space, where the input visual modality is represented in the frequency domain rather than in the spatial domain. We obtain the DCT decomposition by first dividing the visual input into 2D or 3D blocks of pixels of size 4 or 8 and applying DCT to each of them. Different stages are formulated as the progressive modeling of DCT components of increased frequency. We find, however, that DiT models provide reduced performance when working in the frequency rather than the spatial domain, a finding we speculate may originate from an increased difficulty of leveraging token similarities in attention operations. While experiments showed that DCT can be used as a way of producing the input decomposition, input to the DiT model should be converted to the spatial domain for optimal performance, and the Laplacian decomposition is preferred due to its simplicity.

**Alternative parameter specialization methods** As parameter specialization showed the capability of increasing the model's performance (see Table. 1 (d)), we investigate alternative ways of specializing model parameters per each stage. As an alternative avenue, we investigate the recent Tokenformer [47] architecture due to its capacity to progressively integrate new parameters during training. In particular, we instantiate a shared set of parameters and a set of specialized parameters for each stage. The shared set of parameters is always active, while the specialized parameters are activated only when corresponding to the currently active stage. We find the Tokenformer [47]

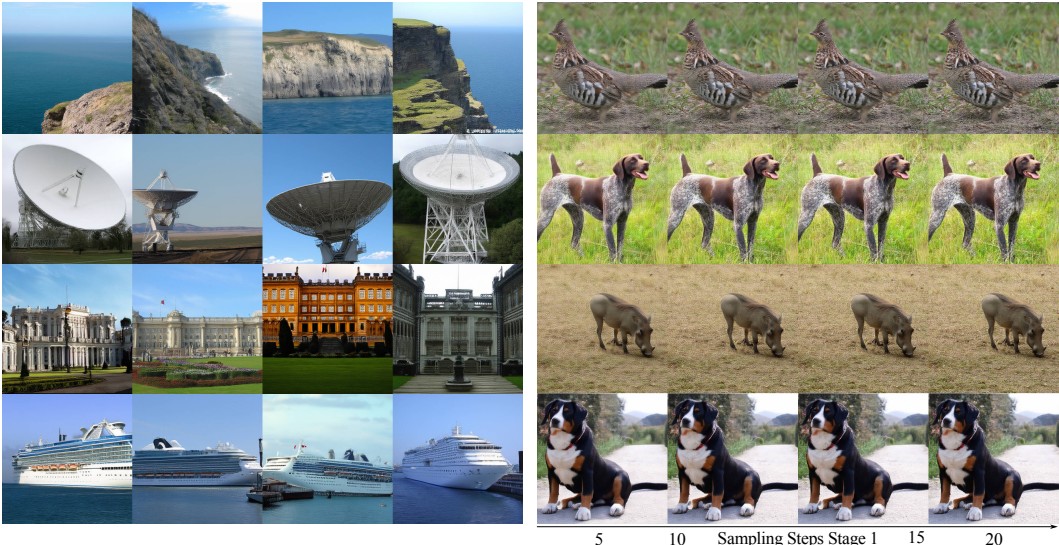

Figure 7: (left) Selected samples highlighting failure cases generated by our framework trained on ImageNet-1K [5] 512px on DiT-XL for 1.3M iterations. When generating high-frequency details such as fine structures, vegetation or cluttered environments, DFM may produce artifacts. (right) Artifacts can be mitigated by increasing the number of sampling steps for the second stage.

architecture to underperform the regular DiT design when given the same amount of computation, thus we abort this experiment. To reduce the number of parameters required for specialization, we wrap each linear layer in different LoRA wrappers [15, 44], one for each stage, and activate the respective wrapper when the corresponding stage is selected at training or inference. LoRA showed improved results over the baseline when utilizing high LoRA ranks which resulted in similar parameter counts to full parameter specialization.

**Expert Models** We train separate or *expert* models for each stage rather than a joint model. We then evaluate performance by mixing expert models trained for different numbers of steps. We find that performance improves steadily with more training of the first stage and improves marginally with more training with longer training of the second stage model, supporting the intuition that modeling of structural detail has a larger importance to sample quality. To reduce the computational burden of training separate models for each stage, we explore finetuning the second stage model starting from the first stage model with positive results. Despite positive results, the idea is not explored further due to the increased complexity of maintaining separate models for each stage. We note that, in principle, it is possible to obtain expert models by finetuning a jointly trained model separately for each stage with full or LoRA finetuning. We leave the exploration of this possibility as future work.

# G   Limitations

DFM improves modeling of visual inputs by decomposing them into different components. The framework's hyperparameters control the amount of model capacity dedicated to modeling each of the components. As an example, a larger probability of sampling stage 0 ($p_t^0$) during training (see Table. 1 (a)) results in a larger emphasis on structural details. As shown in Figure 7, selected samples containing large amounts of high-frequency components such as vegetation, fur, thin structures, or cluttered environments may exhibit artifacts in such regions which manifest as a flattened appearance. Increasing the number of sampling steps for the second stage (see Figure 7 (right) and Figure 18) mitigates such artifacts. By acting on training sampling probabilities for each stage, and distribution of sampling steps between different stages, the framework allows for balance between structural and fine details modeling quality.

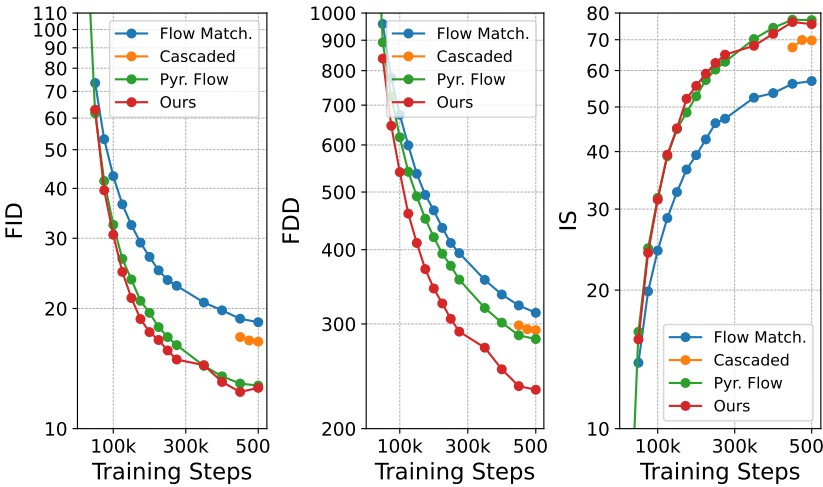

Figure 8: Convergence curves for different datasets and metrics comparing our framework to baselines on ImageNet-1K [5] 512px.

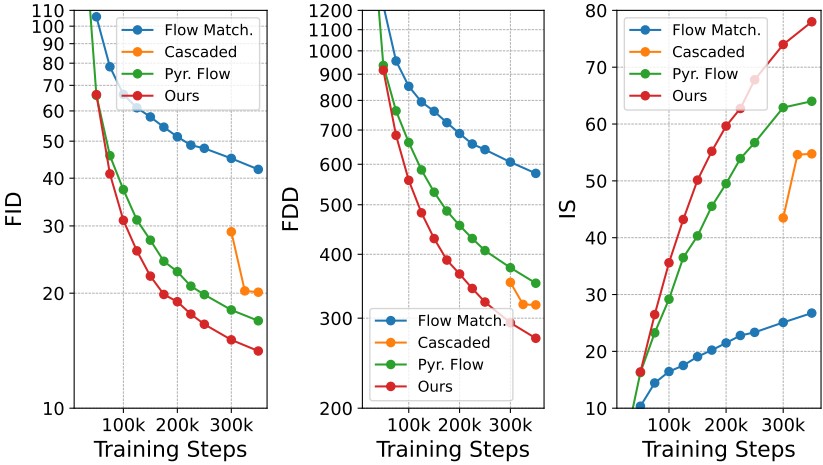

Figure 9: Convergence curves for different datasets and metrics comparing our framework to baselines on ImageNet-1K [5] 1024px.

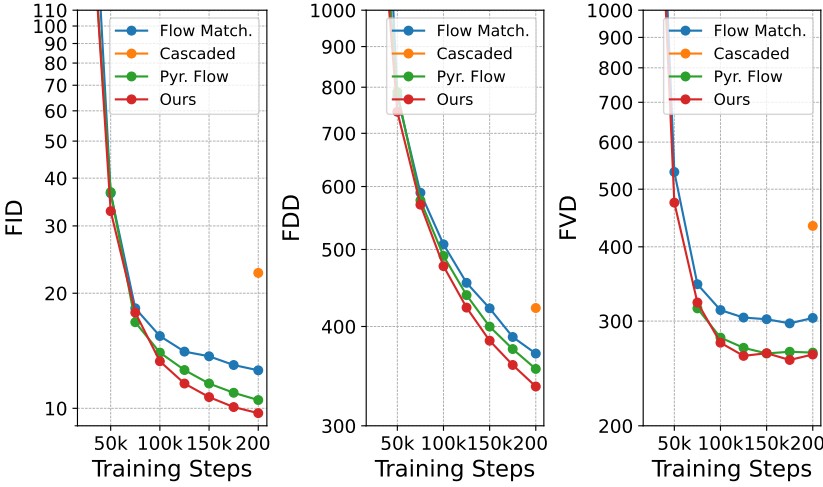

Figure 10: Convergence curves for different datasets and metrics comparing our framework to baselines on Kinetics-700 [3] 512px.

DFM (Ours)   Flow Matching   Cascaded   Pyramidal Flow   DFM (Ours)   Flow Matching   Cascaded   Pyramidal Flow

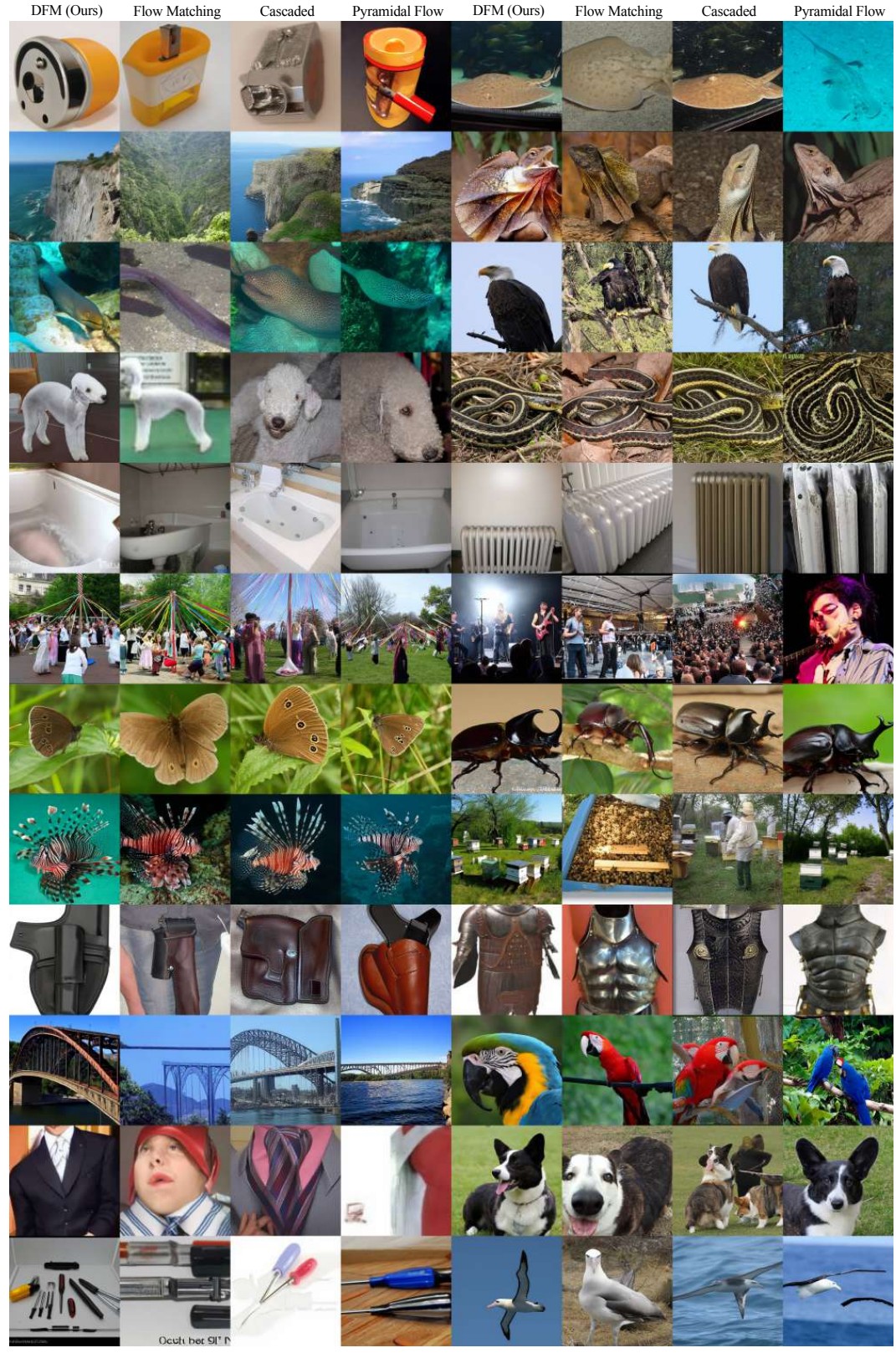

Figure 11: Comparison of DFM against baselines on DiT-XL trained on ImageNet-1K 512px [5] for 500k steps. Samples are fully uncurated and generated with cfg 3.0.

| DFM (Ours) | Flow Matching | Cascaded | Pyramidal Flow | DFM (Ours) | Flow Matching | Cascaded | Pyramidal Flow |

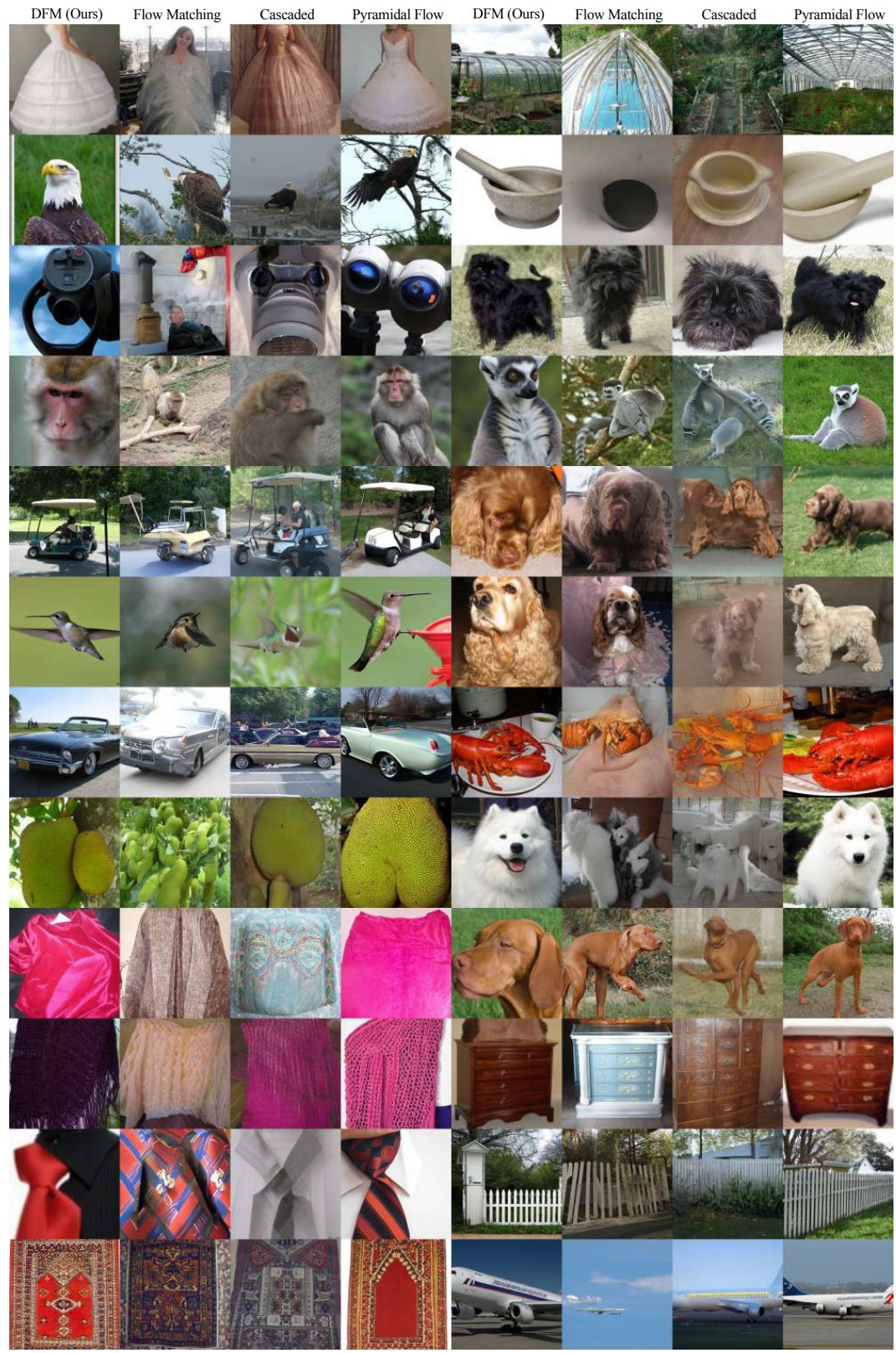

Figure 12: Comparison of DFM against baselines on DiT-XL trained on ImageNet-1K 1024px [5] for 350k steps. Samples are fully uncurated and generated with cfg 3.0.

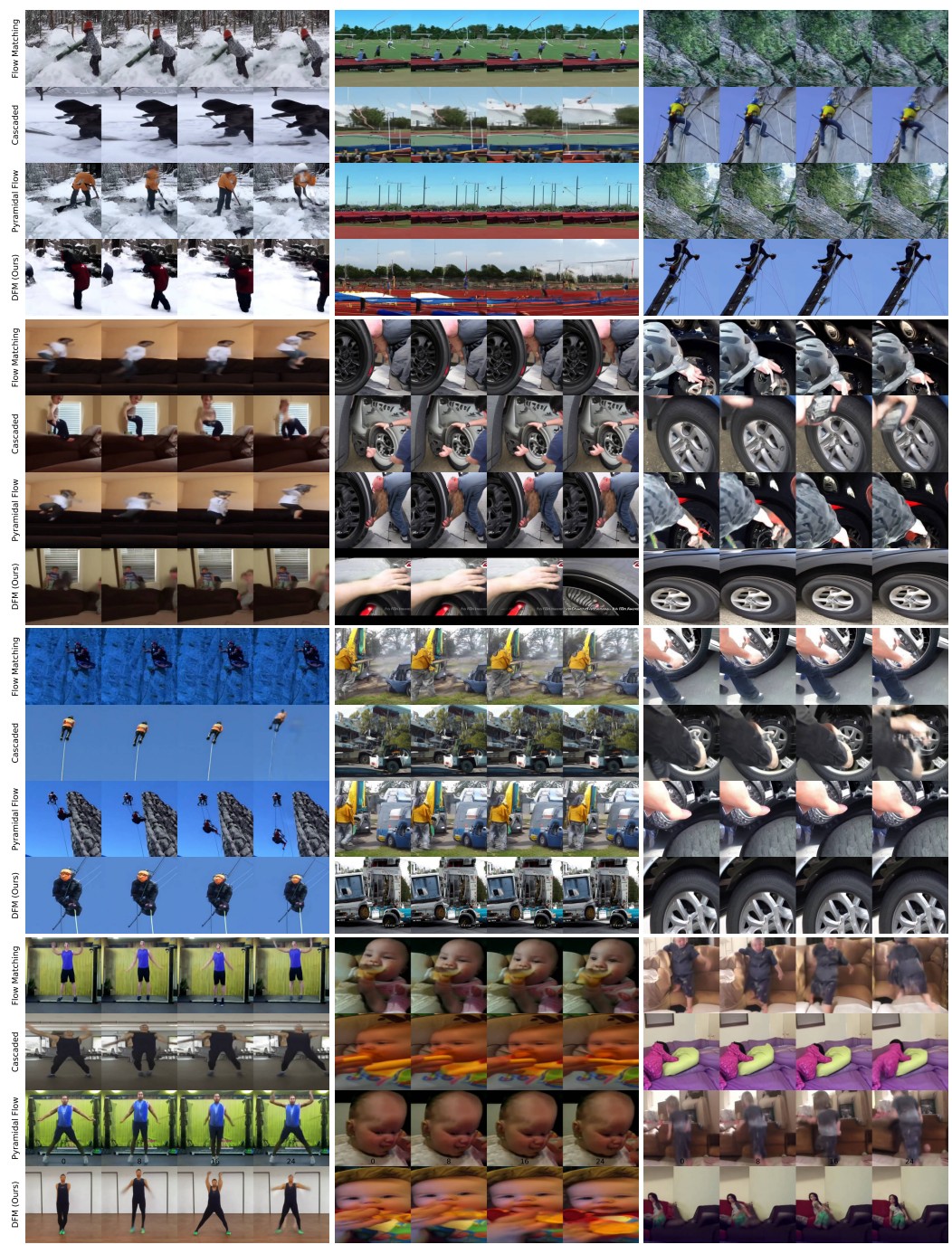

Figure 13: Comparison of DFM against baselines on DiT-XL trained on Kinetics-700 [3] 512px for 200k steps. Samples are fully uncurated and generated with cfg 3.0.

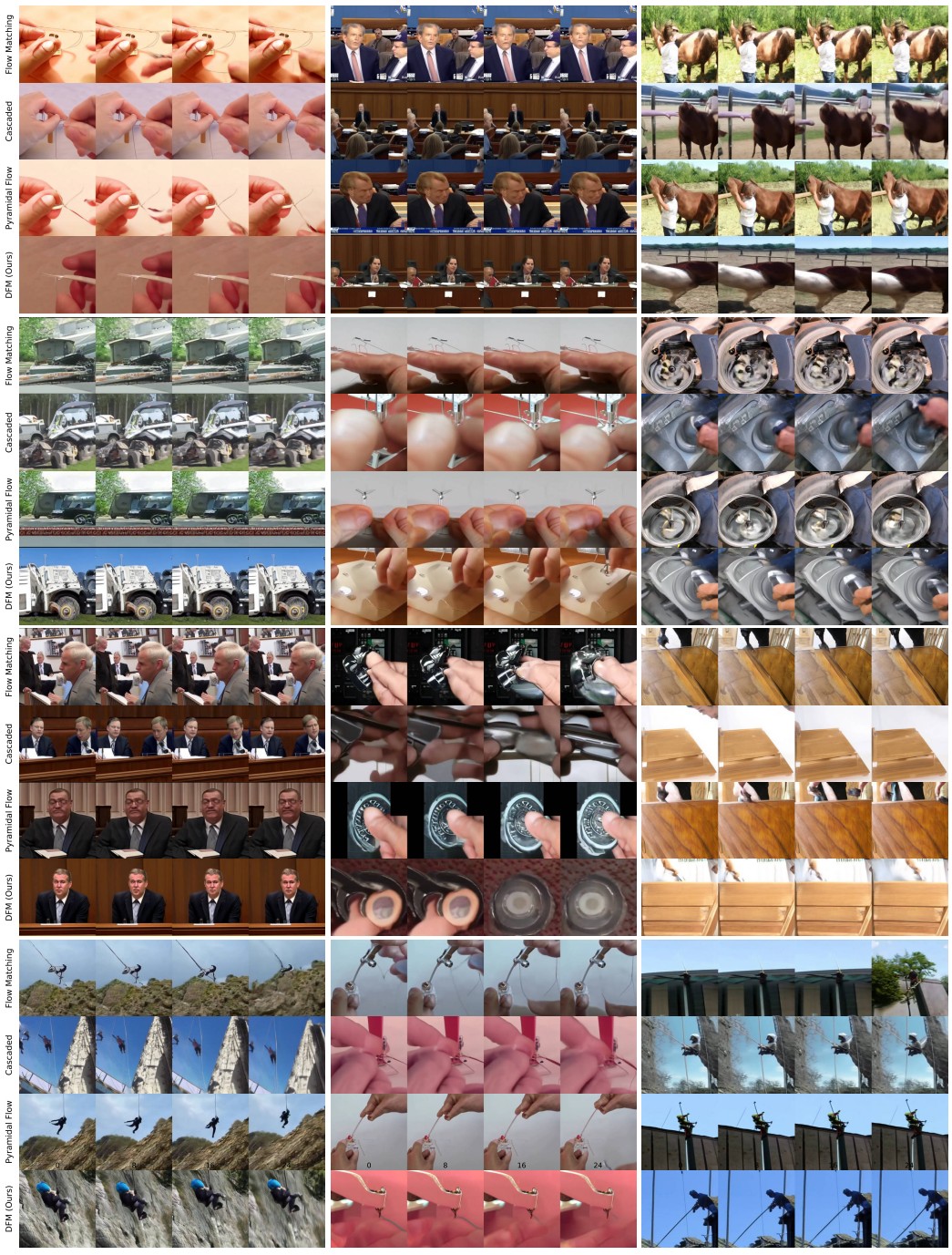

Figure 14: Comparison of DFM against baselines on DiT-XL trained on Kinetics-700 [3] 512px for 200k steps. Samples are fully uncurated and generated with cfg 3.0.

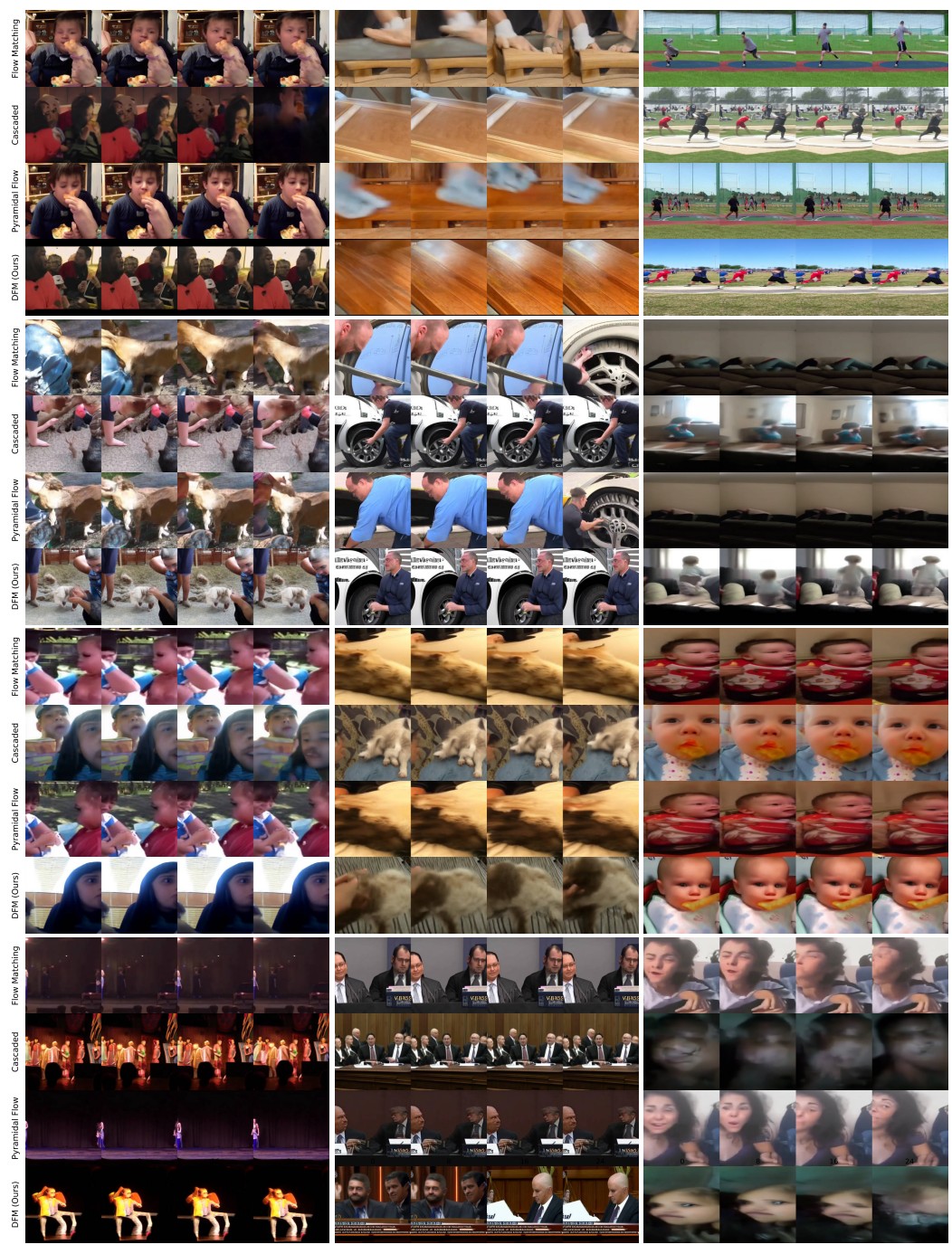

Figure 15: Comparison of DFM against baselines on DiT-XL trained on Kinetics-700 [3] 512px for 200k steps. Samples are fully uncurated and generated with cfg 3.0.

FLUX-DFM  FLUX-FT      FLUX-DFM  FLUX-FT      FLUX-DFM  FLUX-FT

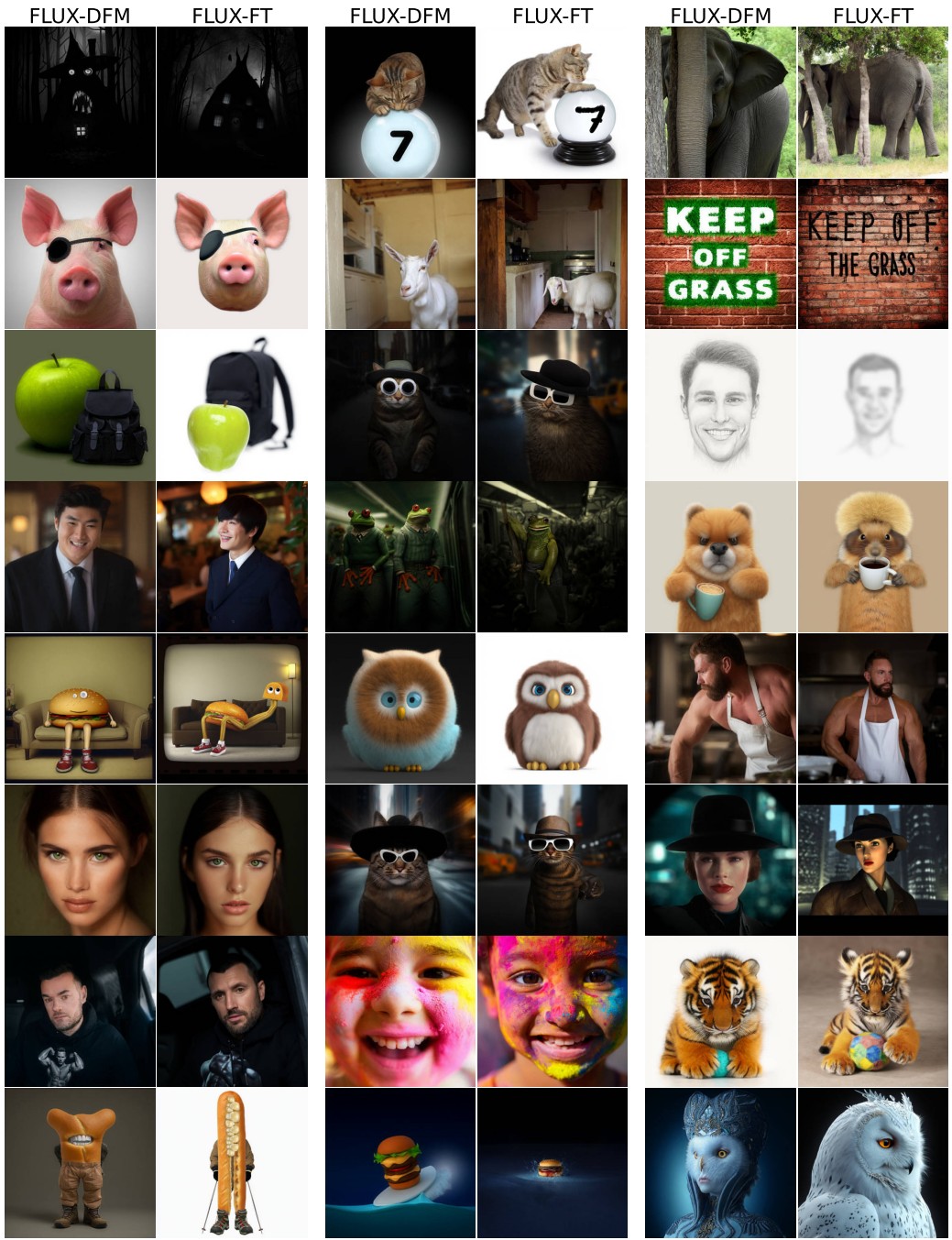

Figure 16: Comparison of finetuning FLUX-DEV with DFM against standard full finetuning trained
finetuned for 24k steps. Samples are generated with cfg 4.5.

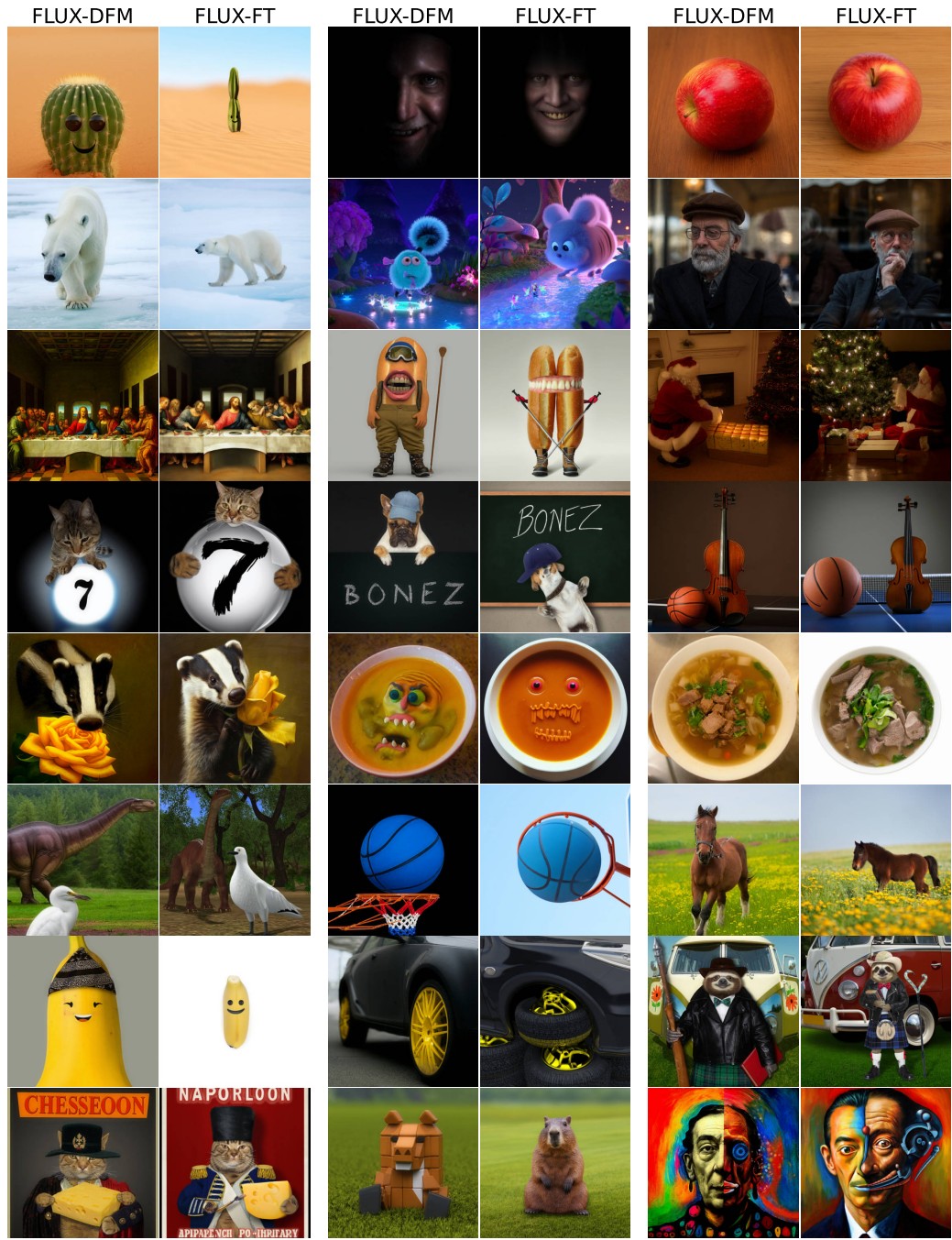

Figure 17: Comparison of finetuning FLUX-DEV with DFM against standard full finetuning trained finetuned for 24k steps. Samples are generated with cfg 4.5.

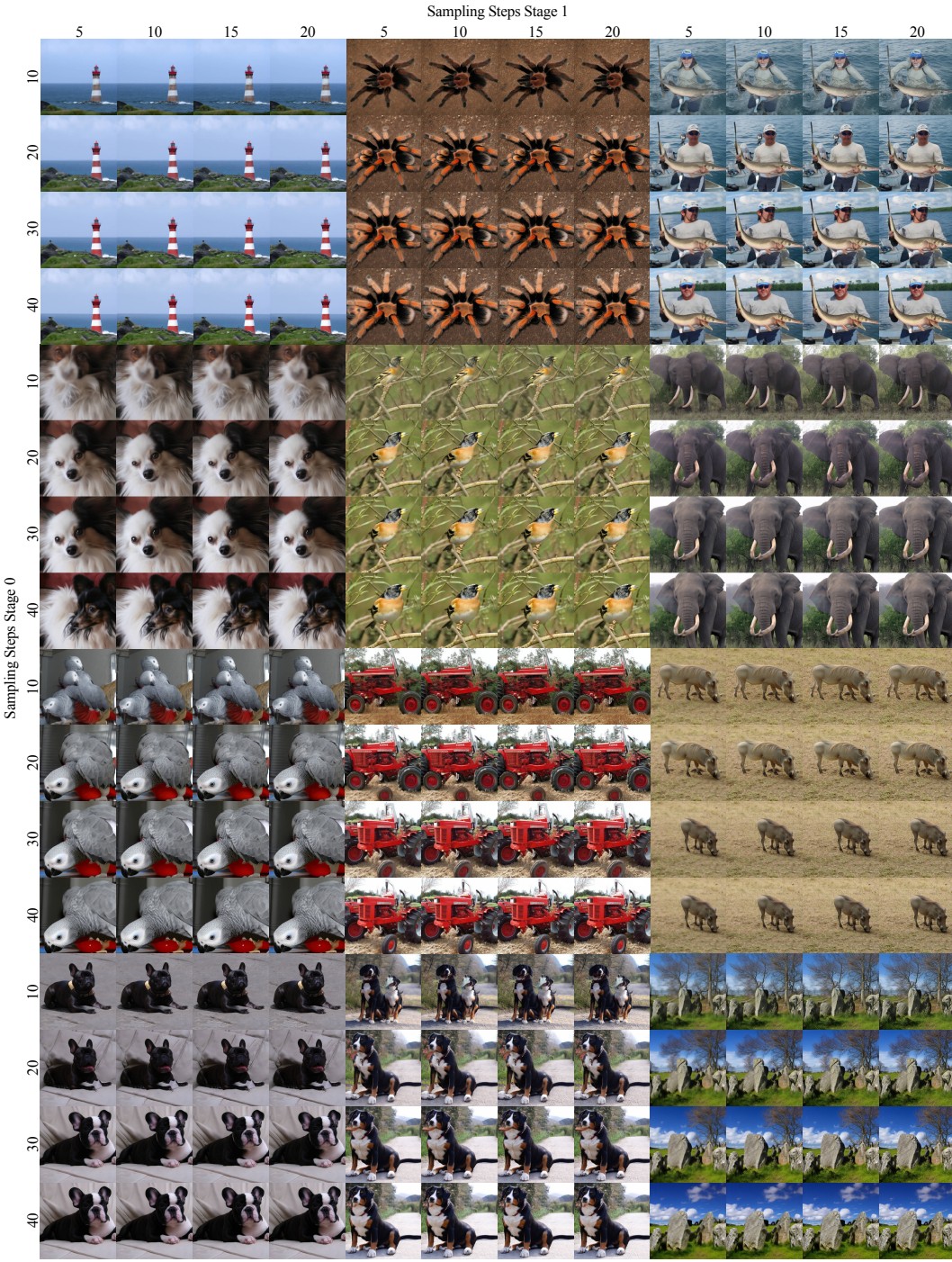

Figure 18: Ablation of per-stage sampling steps on DiT-XL trained on ImageNet-1K 512px [5] for 500k steps. Samples are selected to highlight the effects of varying sampling parameters. Samples are generated with cfg 3.0.

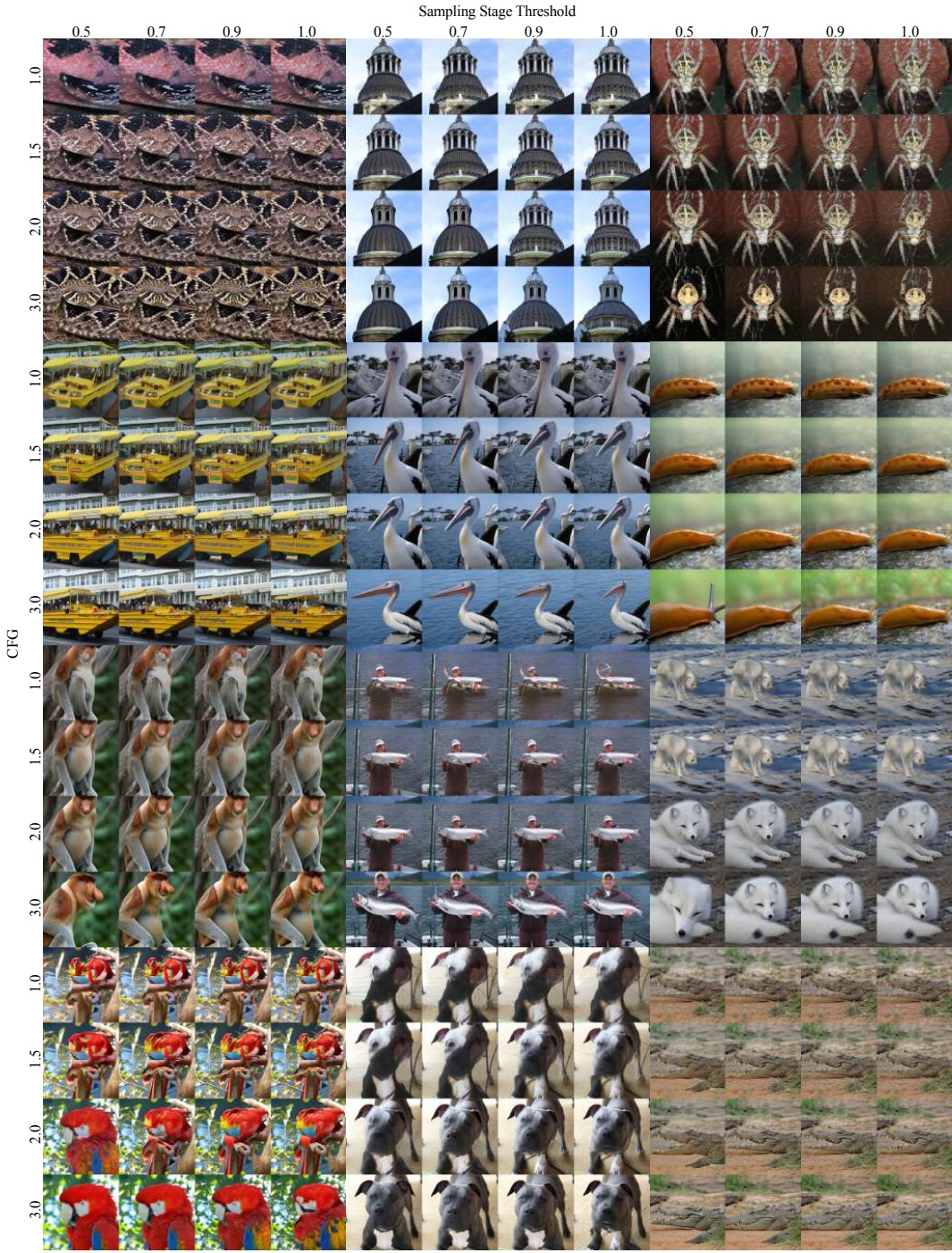

Figure 19: Ablation of the effect of cfg values and sampling threshold $\tau$ on DiT-XL trained on ImageNet-1K 512px [5] for 500k steps. Samples are selected to highlight the effects of varying sampling parameters. Samples are generated with cfg 3.0.

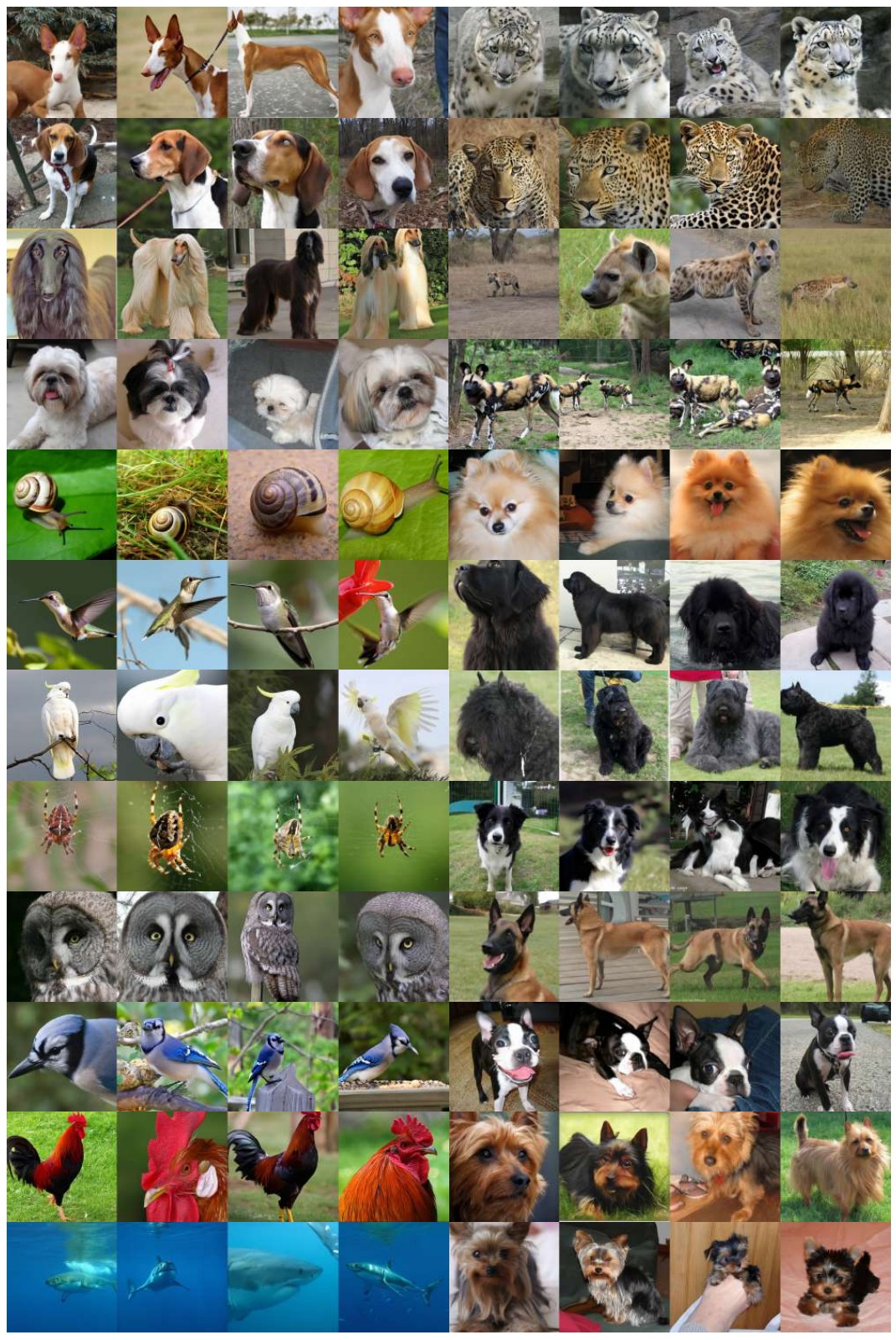

Figure 20: Qualitative results from selected classes on ImageNet-1K [5] 512px produced by DiT-XL trained with DFM for 1.3M steps. We use 30 and 10 sampling steps respectively for the first and second stages and a cfg value of 3.0.

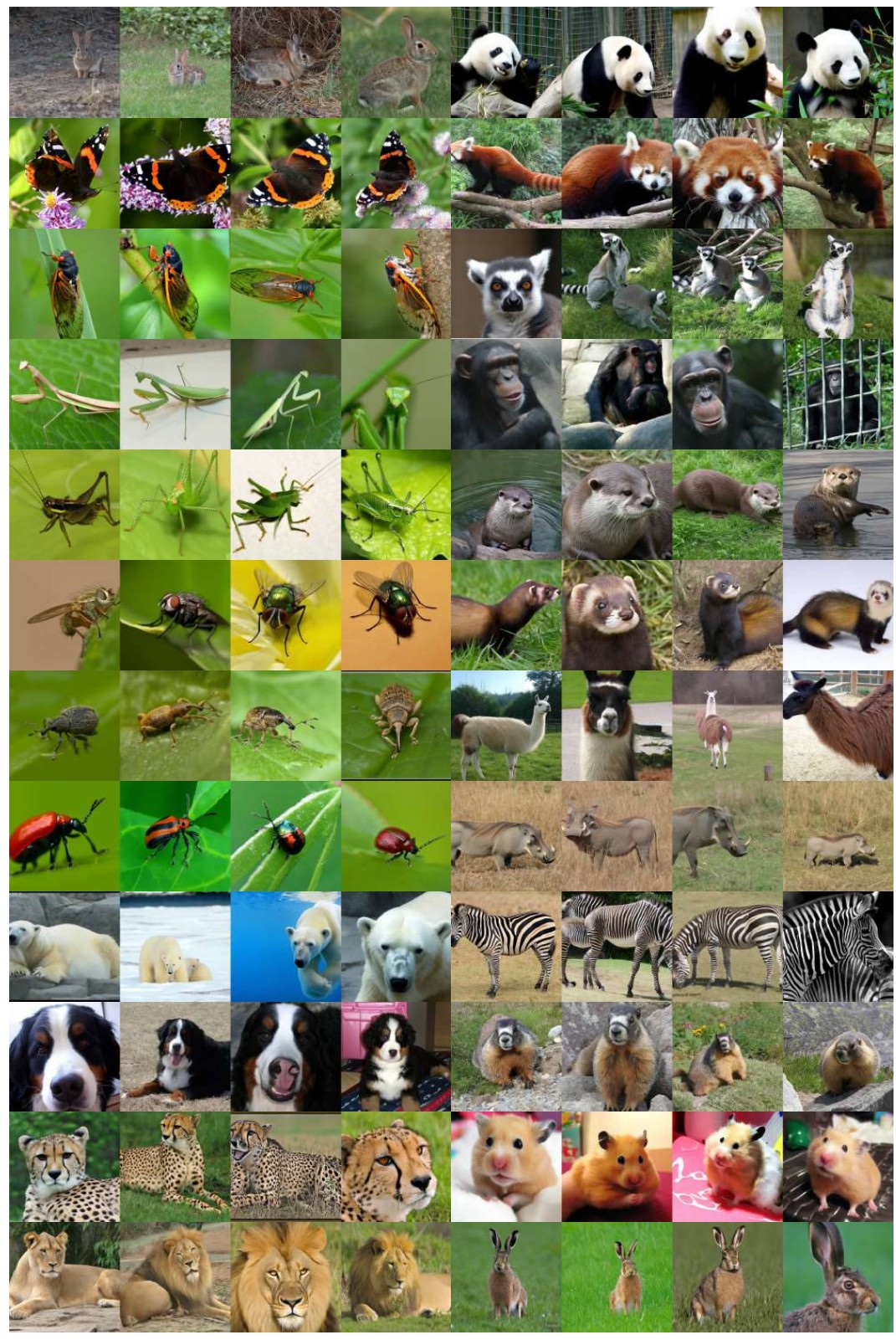

Figure 21: Qualitative results from selected classes on ImageNet-1K [5] 512px produced by DiT-XL trained with DFM for 1.3M steps. We use 30 and 10 sampling steps respectively for the first and second stages and a cfg value of 3.0.

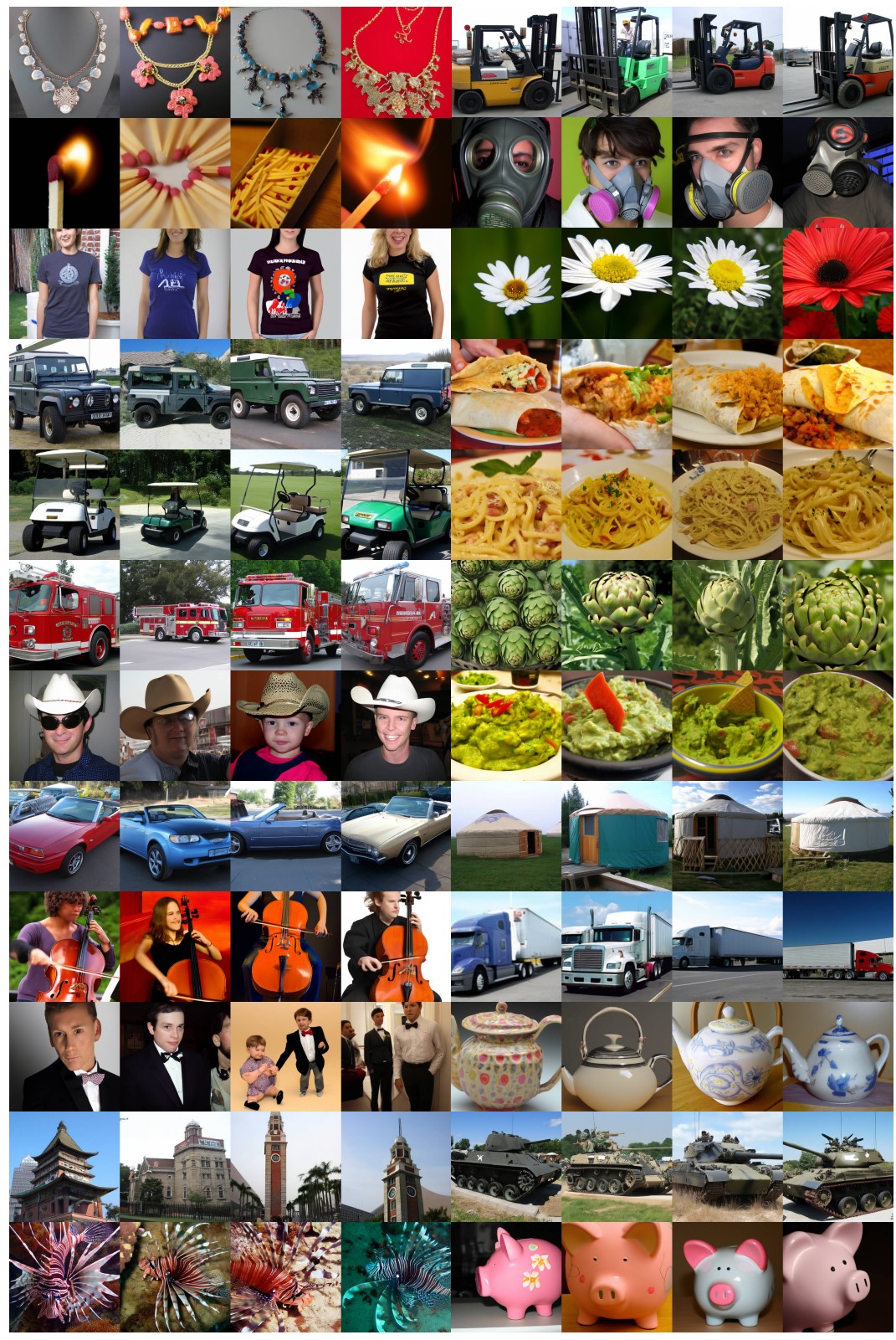

Figure 22: Qualitative results from selected classes on ImageNet-1K [5] 512px produced by DiT-XL trained with DFM for 1.3M steps. We use 30 and 10 sampling steps respectively for the first and second stages and a cfg value of 3.0.

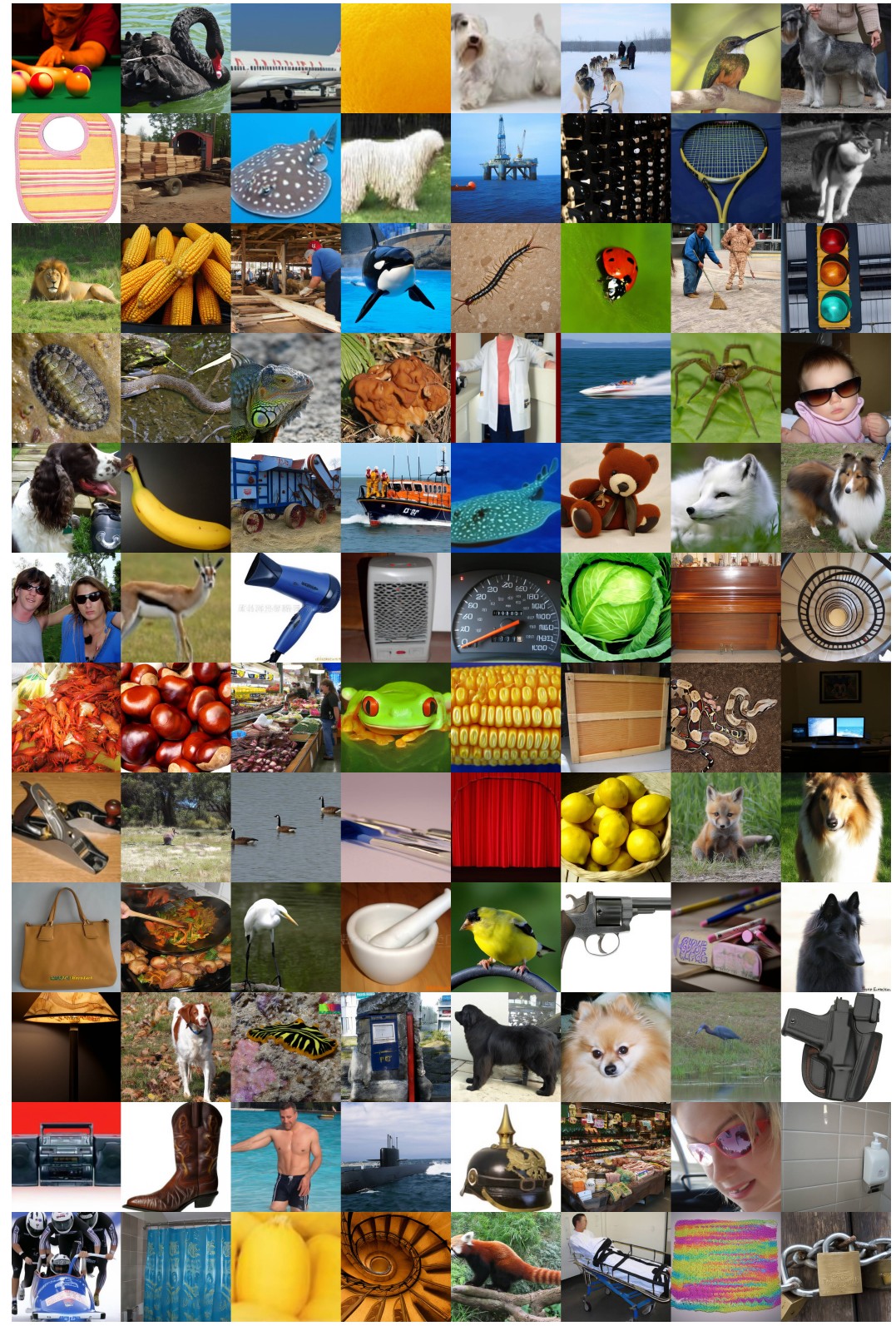

Figure 23: Fully uncurated samples from ImageNet-1K [5] 512px produced by DiT-XL trained with DFM for 1.3M steps. We use 30 and 10 sampling steps respectively for the first and second stages and a cfg value of 3.0.

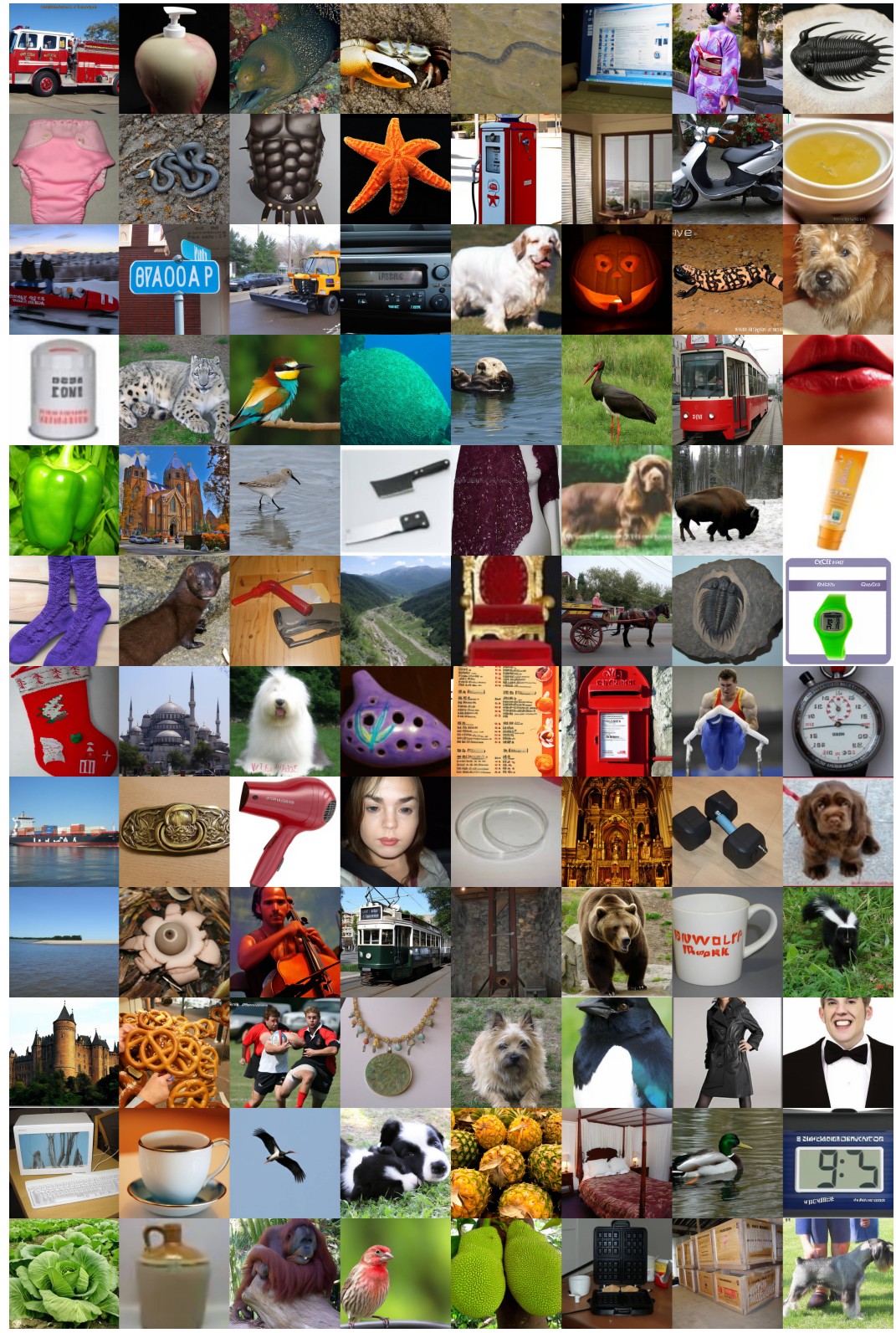

Figure 24: Fully uncurated samples from ImageNet-1K [5] 512px produced by DiT-XL trained with DFM for 1.3M steps. We use 30 and 10 sampling steps respectively for the first and second stages and a cfg value of 3.0.

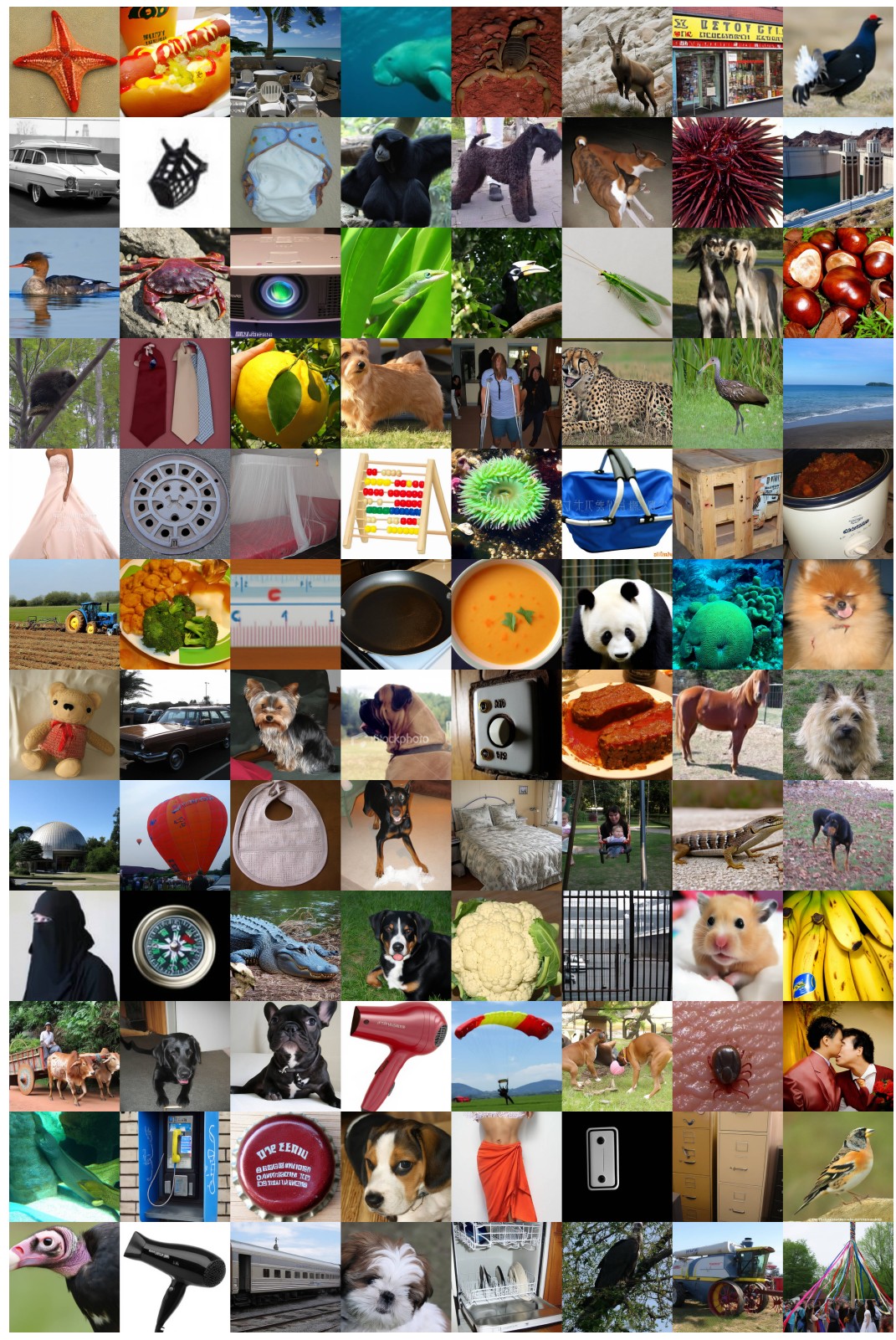

Figure 25: Fully uncurated samples from ImageNet-1K [5] 512px produced by DiT-XL trained with DFM for 1.3M steps. We use 30 and 10 sampling steps respectively for the first and second stages and a cfg value of 3.0.

