# OpenReview forum: "Improving Progressive Generation with Decomposable Flow Matching"
_NeurIPS.cc/2025/Conference — NeurIPS 2025 poster_

### Official Review · Reviewer_uRBr · 2025-06-25

**Clarity:** 3
**Significance:** 3
**Originality:** 3
**Rating:** 4
**Confidence:** 3

**Summary:**

The paper introduces Decomposable Flow Matching (DFM), a novel generative modeling framework that decomposes an input into multiple coarse-to-fine scales and apply flow matching jointly to those components. The authors mainly focused on the Laplacian pyramid and show that DFM reports substantial gains across benchmarks.

**Questions:**

The sampling schedule used at inference time appears interesting. The threshold values (0.7/0.95) suggest that the model spends the majority of its sampling steps on the low-frequency component, with relatively few steps allocated to the high-frequency component. This raises the question of whether performance might improve by fully denoising the low-frequency component first, before proceeding to the high-frequency component, which is similar in spirit to a cascaded model, but with both stages handled by the same model rather than separate ones.

**Ethical Concerns:**

["NO or VERY MINOR ethics concerns only"]

**Final Justification:**

After reading the rebuttal and the other reviewers’ comments, I believe this is a good piece of work, and I lean toward accepting the paper.

**Limitations:**

Yes

**Paper Formatting Concerns:**

Minimal typos (Ln. 7, ad hoc instead of add hoc and Ln. 159, sampled stages)

**Quality:**

3

**Strengths And Weaknesses:**

## Strengths
- The key idea of applying flow matching at multiple scales in a single model is interesting and novel.
- Unlike cascaded models, DFM uses a single network to handle all components of the image.
- The ablation study is well carried out.

## Weaknesses
- The comparative evaluations against baseline models are conducted under a fixed training budget, which is reasonable. However, a deeper analysis of training and inference complexity is necessary. Specifically, DFM’s design feeds all image components into the DiT simultaneously. Although these components have different patch sizes to alleviate the problem, their joint processing could increase memory usage, as each token must attend to all components. For instance, a two-stage DFM would effectively double the number of patches and could potentially quadruple attention computation if I understand correctly, which leads to significantly higher memory requirements and slower training. Could the authors elaborate on that and provide an additional analysis reporting memory usage and training/inference speeds compared to baseline models?

---

> ### Author Rebuttal · Authors · 2025-07-30
>
> We thank all reviewers for their thoughtful feedback. We are encouraged that the paper received positive assessments. Reviewers Nk2B and uRBr described the method as interesting, neat, novel, and original. Reviewer hhY3 found the presentation clear and easy to follow, and Reviewer 6uMG emphasized that the approach is both simple and straightforward to implement. Moreover, reviewers 6uMG and Nk2B found our experiments to be extensive and the method to deliver strong performance.
> Below, we provide detailed responses to Review uRBr questions and concerns.
>
> **Question 1**
> >  Specifically, DFM’s design feeds all image components into the DiT simultaneously. Although these components have different patch sizes to alleviate the problem, their joint processing could increase memory usage, as each token must attend to all components. For instance, a two-stage DFM would effectively double the number of patches and could potentially quadruple attention computation if I understand correctly, which leads to significantly higher memory requirements and slower training. Could the authors elaborate on that and provide an additional analysis reporting memory usage and training/inference speeds compared to baseline models?
>
> We thank the reviewer for their question. DFM aggregates token representations from different scales by summing them before input to the transformer block (Lines 180–181), thus preserving the same memory and compute requirements as the baseline. This is possible since we maintain a consistent number of tokens across scales by adjusting the patch size (L. 176-179). In Table R6, we compare DFM with Flow Matching in terms of peak CUDA memory usage, GFLOPs, forward and training iterations latency, and sampling time (including decoding the image). As shown, DFM introduces only a small overhead. We have included these results in the Appendix.
>
> **Table R6:** Comparison of DFM efficiency compared with Flow Matching measured on 512px Image input and batch size 1.
> | Model          | # Params   | Peak Memory  | GFLOPs  | Forward Latency  | Train Iter Speed  | Sampling Time  |
> |----------------|------------|----------------|-----------|--------------------|---------------------|------------------|
> | Flow Matching  | 712.122 M  | 6.258 GB       | 524.613   | 60.25 ms           | 297.39 ms           | 2.43 s           |
> | DFM            | 716.482 M  | 6.294 GB       | 524.638   | 61.32 ms           | 301.00 ms           | 2.43 s           |
>
>
> > The sampling schedule used at inference time appears interesting. The threshold values (0.7/0.95) suggest that the model spends the majority of its sampling steps on the low-frequency component, with relatively few steps allocated to the high-frequency component. This raises the question of whether performance might improve by fully denoising the low-frequency component first, before proceeding to the high-frequency component, which is similar in spirit to a cascaded model, but with both stages handled by the same model rather than separate ones.
>
> We thank the reviewer for their observation. We have indeed explored fully denoising the first (low-frequency) scale before proceeding with the second (high-frequency) stage, and reported the results in **Figure 3 (a)** (denoted as $\\tau=1.0$). However, we found this setting to underperform compared to $\\tau=0.7$. We hypothesize that $\\tau=1.0$ introduces a degree of train-test mismatch between stages, leading to degraded performance. Such train-test mismatch is alleviated at larger CFG values.
>
> We also note that compared to a cascaded model, a core difference of DFM is that it also restricts the second stage to only generate the high-frequency information, whereas cascaded models generate the full image from scratch in the second stage, conditioned on the first stage output.

---

> > ### Comment · Reviewer_uRBr · 2025-08-04
> >
> > Thank you for the clarification provided in the rebuttal; the paper is now clearer to me. After reading the other reviewers’ assessments, I feel comfortable maintaining my score of borderline accept.

---

> ### Author Response · Authors · 2025-08-05
>
> Dear Reviewer uRBr,
>
> Thank you once again for your thoughtful feedback which helped us improve the clarity of our paper, especially with regard to the method efficiency. We are glad that our responses have addressed your concerns. If you have any additional questions or suggestions, we remain available throughout the discussion period.

---

### Official Review · Reviewer_Nk2B · 2025-06-30

**Clarity:** 1
**Significance:** 3
**Originality:** 4
**Rating:** 5
**Confidence:** 3

**Summary:**

Decomposable Flow Matching (DFM) use a multi-scale learning to learn different resolutions (Laplacian pyramid). Given all noisy images at different scale on different noise levels, predict the per-scale velocities. At inference, progressively generates from low-res to high-res while conditioning on all finished resolutions. They use different patch sizes to handle the multi-scale within a single model. They fine-tune existing image and video autoencoders for any-resolution before training their models.

**Questions:**

- Any intuition why 256->512-> 1024 was worse?
- why logit-normal over uniform for t?

**Ethical Concerns:**

["NO or VERY MINOR ethics concerns only"]

**Final Justification:**

The authors have done extra ablations and reorganized the paper (although I cannot see it) based on my comments to improve the paper. I am satisfied and leaving my score as "accept".

**Limitations:**

Yes, mainly the hyperparameters.

**Paper Formatting Concerns:**

No issues with the guidelines, but formatting in general should be improved.

**Quality:**

3

**Strengths And Weaknesses:**

- Multi-scale progressive growing within a single model is very neat compared to the engineering-heavy cascaded approach.
- FID not FDD in abstract
- The results are strong and the method is much cleaner than cascaded diffusion. This is very good.
- The biggest weakness of the paper is that its not well written, the authors should really make an effort to improve the clarity of the work.
- Its not clear or well explained how the distribution of t is chosen, why is it logit-normal? The visualization from figure 2b time-step sampling should be shown in 4.2.
- Section 4.2 is weird, why are we only learning more about the distribution of t in the experiments sections, when it should be clarified in the methods with a nice figure.
- In the ablations it would be great to see what happens with regular uniform distribution for t
- This work will be high-impact as it makes cascaded diffusion obsolete, since this is much simpler and cleaner training all-in-one with better results

---

> ### Author Rebuttal · Authors · 2025-07-30
>
> We thank all reviewers for their thoughtful feedback. We are encouraged that the paper received positive assessments. Reviewers Nk2B and uRBr described the method as interesting, neat, novel, and original. Reviewer hhY3 found the presentation clear and easy to follow, and Reviewer 6uMG emphasized that the approach is both simple and straightforward to implement. Moreover, reviewers 6uMG and Nk2B found our experiments to be extensive and the method to deliver strong performance.
> Below, we provide detailed responses to Review Nk2B questions and concerns.
>
> **Question 1**
> > FID not FDD in abstract
>
> We refer to Frechet DINOv2 Distance (FDD) in the abstract. On ImageNet 512px with CFG=1.5, the base architecture (Flow Matching) achieves an FDD of 132.5 while DFM reports 85.8, marking 35.2% relative improvement. We have revised the abstract to spell out the full metric name and specify the CFG value used for these results.
>
>
> **Question 2**
> > Its not clear or well explained how the distribution of t is chosen, why is it logit-normal?
>
> > In the ablations it would be great to see what happens with regular uniform distribution for t
>
> > why logit-normal over uniform for t?
>
> The logit-normal timestep distribution was first introduced in Stable Diffusion 3 [1], and its effectiveness has been verified in following work [2]. It has become a standard approach for timestep sampling in the latest diffusion-based methods [3-6]. Motivated by this, we adopted the logit-normal distribution throughout our paper. As suggested, we conducted additional experiments replacing logit-normal sampling with uniform sampling for both the Flow Matching baseline and DFM. As reported in Table R4, we observe a consistent drop in performance for both models, confirming the advantage of the logit-normal distribution. We will include these results in the supplementary material.
>
> **Table R4:** Ablations of DFM and Flow Matching with uniform and logit-normal timestep distribution, trained on ImageNet 512px with a DiT-B/2 backbone.
> | Baseline                     | FID ↓   | FDD ↓     | IS ↑    |
> |-----------------------------|---------|-----------|---------|
> | Flow Matching w/ logit-normal | 38.01   | 665.16    | 30.52   |
> | Flow Matching w/ uniform      | 45.61   | 834.92    | 26.48   |
> | DFM w/ logit-normal           | 29.83   | 567.21    | 40.38   |
> | DFM w/ uniform                | 32.87   | 651.09    | 36.23   |
>
> **Question 3**
> > The biggest weakness of the paper is that its not well written, the authors should really make an effort to improve the clarity of the work.
>
> We thank the Reviewer for their feedback. We understand from the Reviewer’s punctual comments that the source of the clarity issues resides in the discussion of the adopted timestep distributions. Below, we include clarifications on the timestep sampling.
>
> **Timestep Sampling**
>
> DFM operates on different stages, where each stage contains multi-scale representations. For instance, for the first stage modeling, only the first (low-frequency) scale is modeled, while in the second stage, both scales (low and high frequency) are jointly modeled. During training, we first sample the stage number from a discrete probability distribution (L. 158). Then, we sample the timesteps of different scales independently (L. 140), where each scale’s timestep is drawn from a predetermined logit-normal distribution, parametrized by a location ($ m_s $), where s is the scale index within the sampled stage. We ablated over the choice of $m_s$ in Table. 2 (b), and found that it is beneficial to have the distribution of the last scale to use location ($m_s=0$), aligning with common practice in flow matching [1-6], while shifting the location of earlier scales toward lower noisy levels (higher $m_s$ value). This better simulates the inference-time behavior, where during the generation of scale s, the previous scale s−1 has already undergone significant denoising from previous stages. Please see Section 4.2 ("Training timestep distribution") for more details.
>
> We have revised Eq. (4) to better show that timesteps are sampled independently and that the timestep distribution depends on the stage number and scale index within that stage.
>
> $$
> \\mathcal{L}
>   = E_{Q\\sim p_{q},\\; t^{1}\\sim p_{t, Q}^{1},\\dots,t^{S}\\sim p_{t, Q}^{S},\\; X_{1}\\sim p_{d},\\; X_{0}\\sim p_{n}}
>     \\sum_{s=1}^{S}
>       M^{s}\\,
>       \\left\\lVert
>         \\mathcal{G}\\bigl(P_{t^{1},\\dots,t^{S}},\\,t^{1},\\dots,t^{S}\\bigr)
>         - v^{s}
>       \\right\\rVert_{2}^{2}\\,.
> $$
>
> Here, we introduced a new notation Q that corresponds to the stage number (corresponding to the number of modeled scales) for the current training iteration, and revised the discrete distribution notation to $p_q$. $p_{t, Q}^s$ denotes the timestep distribution of the scale s within stage Q. In DFM, for the two scales case ($Q=2$), $p_{t,Q}^1$ follows a logit-normal distribution with location $m_{1} = 0 $ and $p_{t,Q}^0$ follows a logit-normal distribution with location $m_{0} = 1.5$.
>
> Below, we include a pseudo code of the time step sampling, which we will include in the paper for better clarity.
>
> $$
> \\begin{array}{l}
> \\mathrm{procedure}\\;SampleTrainingTimesteps(S):\\\\
> \\mathrm{inputs:}\\;S \\text{    ; total number of scales}\\\\
> \\mathrm{outputs:}\\;(t^{1},\\ldots,t^{S})\\\\[6pt]
> Q\\leftarrow\\mathrm{SampleDiscreteDistribution}(p_{q}) \\text{ ; stage index corresponding to number of modeled scales}\\\\
> t^{Q}\\leftarrow\\mathrm{SampleLogitNormal}(m=0)\\\\
> \\mathrm{for}\\;s=1\\;\\mathrm{to}\\;Q-1:\\\\
> \\quad t^{s}\\leftarrow\\mathrm{SampleLogitNormal}(m=1.5)\\\\
> \\mathrm{end\\;for}\\\\
> \\mathrm{for}\\;s=Q+1\\;\\mathrm{to}\\;S:\\\\
> \\quad t^{s}\\leftarrow 0\\\\
> \\mathrm{end\\;for}\\\\
> \\mathrm{return}\\;(t^{1},\\ldots,t^{S})
> \\end{array}
> $$
>
> We have also implemented paper modifications according to each of the Reviewer’s punctual comments and remain available to further modify the manuscript in case this does not fully address the clarity issues.
>
> **Question 4**
> > The visualization from figure 2b time-step sampling should be shown in 4.2.
>
> As per the Reviewer's suggestion, we improved the clarity of the discussion in Sec 4.2 and supplement it with an inset figure visualizing the training timestep distribution of each scale in every stage and the effects of changing the distribution location hyperparameter. Due to NeurIPS rebuttal rules, however, we are unable to attach the new figure here.
>
> **Question 5**
> > Section 4.2 is weird, why are we only learning more about the distribution of t in the experiments sections, when it should be clarified in the methods with a nice figure.
>
> We thank the Reviewer for the suggestion. We introduced **in the method section (3.3)**, the inference timestep scheduler (L. 148-154) and the training timestep distribution (L. 155-163), where we explained that after sampling the stage number according to a discrete probability distribution, the timesteps of each scale are sampled independently. The timestep corresponding to the last scale of each stage follows a logit-normal distribution, while the timestep corresponding to previous scales follows the same logit-normal distribution but shifted towards lower noise levels. Since the amount of shifting is a hyperparameter derived empirically, we discuss its value in the experiments section to support it with an ablation study (Tab. 1 (b)). Following the Reviewer’s suggestion, we supplemented Section 4.2 with a figure showing the exact distribution of the training timestep of each scale in every stage in accordance with the ablated “location” hyper-parameter.
>
> **Question 6**
> > Any intuition why 256->512-> 1024 was worse?
>
> We hypothesize that the performance gap between the “256->512->1024” and “256->1024” experiment in Tab. 2 (f) comes from differences in stage sampling probabilities during training $p_q$ (as ablated in Tab. 1 (a) for the 2 stages setting). In the ablation experiment of Tab. 1 (f), stage sampling probabilities are set as follows:
> - 512->1024: (0.9, 0.1)
> - 256->512->1024: (0.8, 0.1, 0.1)
> - 256->1024: (0.9, 0.1)
>
> The (0.8, 0.1, 0.1) choice for the “256->512->1024” experiment was operated to match the “512->1024” experiment's second-stage sampling probabilities, with the effect of undermining low-frequency training.
> To verify this hypothesis, we repeat the experiment, setting the “256->512->1024” stage sampling probabilities during training to (0.9, 0.05, 0.05), thus matching the best-performing “256->1024” experiment in the low-frequency training. At inference time, we also vary sampling steps per stage from (20, 12, 8) respectively for each stage, to (25, 10, 5), bringing it closer to the 2-stage setting of (25, 15). As shown in Table R5, the performance gap between “256->512->1024” and “256->1024” is no longer present, and 3 stages deliver slightly improved performance.
>
> **Table R5:** Ablations of three-stage DFM, trained on ImageNet 1024px with a DiT-B/2 backbone. Numbers correspond to CFG values of 1.0.
> | Baseline                         | FID ↓   | FDD ↓   | IS ↑    |
> |----------------------------------|---------|---------|---------|
> | 2 stages                         | 41.06   | 704.0   | 30.9    |
> | 3 stages (0.8, 0.1, 0.1)         | 50.61   | 827.0   | 24.5    |
> | 3 stages (0.9, 0.05, 0.05)       | 40.6    | 683.2   | 31.82   |
>
> **References**
>
> [1] Scaling Rectified Flow Transformers for High-Resolution Image Synthesis, ICML 2024
>
> [2] FasterDiT: Towards Faster Diffusion Transformers Training without Architecture Modification, NeurIPS 2024
>
> [3] SnapGen: Taming High-Resolution Text-to-Image Models for Mobile Devices with Efficient Architectures and Training, CVPR 2025
>
> [4] Wan: Open and Advanced Large-Scale Video Generative Models, 2025
>
> [5] Movie Gen: A Cast of Media Foundation Models, 2024
>
> [6] Seedance 1.0: Exploring the Boundaries of Video Generation Models, 2025

---

> > ### Comment · Reviewer_Nk2B · 2025-08-04
> > **Response**
> >
> > The authors have done extra ablations and reorganized the paper (although I cannot see it) based on my comments to improve the paper. I am satisfied and leaving my score as "accept".

---

### Official Review · Reviewer_6uMG · 2025-07-01

**Clarity:** 3
**Significance:** 3
**Originality:** 2
**Rating:** 4
**Confidence:** 4

**Summary:**

In this paper, the authors propose a progressive generation framework for high-dimension images and videos. They use a Laplacian decomposition strategy to produce a multiscale input representation. The proposed Decomposable Flow Matching (DFM) applies flow matching independently at each level of a user-defined multi-scale representation. In addition, extensive experiments show the effectiveness of the proposed method across various tasks.

**Questions:**

See Weakness.

**Ethical Concerns:**

["NO or VERY MINOR ethics concerns only"]

**Final Justification:**

While my concerns have been adequately addressed, I still find the overall contribution to be borderline in terms of novelty and impact. Therefore, I maintain my score at Borderline Accept.

**Limitations:**

yes

**Quality:**

2

**Strengths And Weaknesses:**

Strengths
+ The proposed method is simple and easy to implement. It is agnostic to the choice of decompositions and does not need multiple models or per-stage training.
+ The experiments is extensive and the performance gain is large. This paper provides a comprehensive analysis of critical training and sampling hyperparameters. In addition, the proposed method outperforms existing sota methods across various tasks.

Weaknesses
- For latent space, we re-train all autoencoders by scale-equivariant finetuning. To my knowledge, this can boost the generation performance. Regarding the baselines in Section 4.4, do they also employ scale-equivariant autoencoders? If not, the comparison may be unfair.
- lognorm location in Table 1 (b). First, there appears to be a potential typo: “second stage” should likely be “first stage.” Second, the main text states that a location value of 1.5 is selected. However, according to Table 1(b), this setting does not yield the best performance. What is the rationale behind choosing 1.5?

---

> ### Author Rebuttal · Authors · 2025-07-30
>
> We thank all reviewers for their thoughtful feedback. We are encouraged that the paper received positive assessments. Reviewers Nk2B and uRBr described the method as interesting, neat, novel, and original. Reviewer hhY3 found the presentation clear and easy to follow, and Reviewer 6uMG emphasized that the approach is both simple and straightforward to implement. Moreover, reviewers 6uMG and Nk2B found our experiments to be extensive and the method to deliver strong performance.
> Below, we provide detailed responses to Review 6uMG questions and concerns.
>
> **Question 1**
> > For latent space, we re-train all autoencoders by scale-equivariant finetuning. To my knowledge, this can boost the generation performance. Regarding the baselines in Section 4.4, do they also employ scale-equivariant autoencoders? If not, the comparison may be unfair.
>
> We confirm that all baselines in Section 4.4 employ the **same scale-equivariant autoencoders**.  We additionally provide in **Appendix. D** a comparison of our method against vanilla Flow Matching, where both use the **standard FLUX autoencoder** (i.e. without scale-equivariance finetunning) and show an even wider margin of improvement for our method. To further demonstrate the generality of DFM across different autoencoder representations, we re-train DFM and Flow Matching using the same settings as in Table. 2 but using the **standard Cogvideo autoencoder** (i.e. without scale-equanariance finetunning). As reported in Table. R3, DFM continues to show similar improvement over Flow Matching. We revised section 4.1 to make it clear that all baselines employ the same autoencoder.
>
> **Table R3:** Quantitative comparison of DFM with CogVideo AE, trained on ImageNet 512px with a DiT-XL/2 backbone. Numbers correspond to CFG values of 1.0, 1.25 and 1.5.
>
> | Experiment             | FID ↓(1.0 / 1.25 / 1.5)     | FDD ↓ (1.0 / 1.25 / 1.5)           | IS ↑ (1.0 / 1.25 / 1.5)             |
> |------------------------|----------------------------|----------------------------------|-----------------------------------|
> | FlowMatch (CogVideo AE)| 16.94 / 8.59 / 5.84        | 319.88 / 218.48 / 159.51         | 58.56 / 96.08 / 137.39            |
> | DFM (CogVideo AE)      | 8.80 / 4.36 / 5.14         | 208.4 / 125.73 / 90.22           | 91.79 / 158.80 / 222.09           |
>
> **Question 2**
> > lognorm location in Table 1 (b). First, there appears to be a potential typo: “second stage” should likely be “first stage.”
>
> We thank the reviewer for pointing this out. During the first stage, only the first scale (low-frequency) is generated. During the second stage, two scales are generated simultaneously: the previous (low-frequency scale), which will be close to being fully denoised, and the current (high-frequency scale). In Table 1 (b), we are referring to the first scale timestep distribution during second stage training. We have revised the table caption to “Previous scale lognorm location” and clarified in Sec. 4.2 the intended meaning. Please refer to our answer to Reviewer hhY3 Question 3 for more general clarifications regarding timestep sampling
>
> **Question 3**
> > Second, the main text states that a location value of 1.5 is selected. However, according to Table 1(b), this setting does not yield the best performance. What is the rationale behind choosing 1.5?
>
> We found that location values of 0.5, 1.0, and 1.5 yield comparable performance, with differences of less than **1% in FDD** and **1.3% in IS** (as reported in Table 1 (b)) when ablations are conducted on DiT/B. A larger location value, however, simulates availability of a cleaner first scale (low-frequency) result and aligns more closely to inference time behavior, where the second generation stage is started once the first stage is close to being fully denoised. We hypothesize that under the increased DiT/XL capacity used in the main experiments, a closer alignment to inference-time behavior would be beneficial. Since Table 1(b) suggests that any location value within the range 0.5–1.5 yields similarly strong results, we prefer choosing 1.5. We briefly discussed this in **Lines 226–228** and have now revised the text in that section to better clarify our rationale.

---

> > ### Author Response · Authors · 2025-08-05
> >
> > Dear Reviewer 6uMG,
> >
> > Thank you once again for your valuable comments and feedback. With just two days remaining in the discussion period, we would like to check whether our answers clarified your concerns. We remain fully available to address any additional questions or concerns you may have.

---

> > ### Comment · Reviewer_6uMG · 2025-08-06
> >
> > I appreciate the authors’ thorough response and the effort they put into the rebuttal. While my concerns have been adequately addressed, I still find the overall contribution to be borderline in terms of novelty and impact. Therefore, I maintain my score at Borderline Accept.

---

### Official Review · Reviewer_hhY3 · 2025-07-02

**Clarity:** 2
**Significance:** 3
**Originality:** 2
**Rating:** 4
**Confidence:** 3

**Summary:**

This paper proposes a progressive generative modeling framework on top of flow matching. The approach first applies a user-defined decomposition to the input image and produces multi-scale representations. Then the flow matching is used to model individual scale representation independently. The sampling procedure generates images progressively by traversing through different scale predictions. The paper claims that the proposed approach avoids requiring expert models or ad hoc designs and improves image and video generation performance significantly.

**Questions:**

I don't follow progressive modeling literatures and would like to hear more from other reviewers regarding novelty. I am open to adjust rating.

**Ethical Concerns:**

["NO or VERY MINOR ethics concerns only"]

**Final Justification:**

The additional experimental results provided by the authors are appreciated. I maintain my rating and lean toward acceptance.

**Limitations:**

yes

**Quality:**

3

**Strengths And Weaknesses:**

**Strength**

1. The writing is smooth and easy to follow.

2. The paper is tackling an interesting problem by decomposing diffusion generation tasks into simpler subtasks through progressive modeling.

**Weakness**
1. Through the paper advocates that the proposed framework avoids the need for custom diffusion formulation, ad-hoc sampler, etc., it highly relies on properly defined multi-scale representations. The paper puts that various techniques can be used like DWT, DCT, or Fourier transforms, while how to determine the number of scales and the size of scales for these approaches is unclear. No principled approach is provided.
2. The paper claims that the progressive modeling achieves better generation performance than Flow Matching. Does this improvement come with greater complexity for training and inference? The framework needs to model muti-scale distributions during training and the sampling also progressively traverses multiple stages. During training, do you sample time steps for different scales jointly? Does that need a lot of samples to approximate the $p_t$ expectation?
3. Typos: add-hoc samplers (in Abstract), and the formulate the .... (line 140), etc.

---

> ### Author Rebuttal · Authors · 2025-07-30
>
> We thank all reviewers for their thoughtful feedback. We are encouraged that the paper received positive assessments. Reviewers Nk2B and uRBr described the method as interesting, neat, novel, and original. Reviewer hhY3 found the presentation clear and easy to follow, and Reviewer 6uMG emphasized that the approach is both simple and straightforward to implement. Moreover, reviewers 6uMG and Nk2B found our experiments to be extensive and the method to deliver strong performance.
> Below, we provide detailed responses to Review hhY3 questions and concerns.
>
> **Question 1**
>
> >Through the paper advocates that the proposed framework avoids the need for custom diffusion formulation, ad-hoc sampler, etc., it highly relies on properly defined multi-scale representations. The paper puts that various techniques can be used like DWT, DCT, or Fourier transforms, while how to determine the number of scales and the size of scales for these approaches is unclear. No principled approach is provided.
>
> Following the Reviewer’s comment, we validate robustness of our suggested Laplacian hyperparameters to diverse decompositions and autoencoders.
>
> DMF relies on a multi-scale spatial representation that separates low- and high-frequency information. Such separation can be achieved with several decomposition methods such as Laplacian, DCT, DWT, etc, by decoding the low- and high-frequency components separately.  Despite their representational differences, we find the practical guidance on choosing the hyper-parameters in Appendix A.5 still applies. We retrained DFM with DCT‑ and DWT‑based decompositions using exactly the same settings as in Table 2. As reported in Table R1, these identical hyper‑parameters deliver the same gains over the Flow‑Matching baseline for every decomposition. We further validate that these conclusions hold irrespective of the autoencoder representation. When we adopt the CogVideo autoencoder and train with identical hyper-parameters, DFM equipped with Laplacian, DCT, or DWT outperforms the baseline by a wide margin (Table R2). We have updated the paper to include these results and expand the discussion on the choice of the decomposition method.
>
> **Table R1:** Quantitative comparison of DFM equipped with various decomposition methods, trained on ImageNet 512px with a DiT-XL/2 backbone and scale-equavariant FLUX AE. Numbers correspond to CFG values of 1.0, 1.25 and 1.5.
>
> | Experiment        | FID_50k (1.0 / 1.25 / 1.5)    | FDD_50k (1.0 / 1.25 / 1.5)       | IS (1.0 / 1.25 / 1.5)             |
> |------------------|-------------------------------|----------------------------------|-----------------------------------|
> | Flow Matching     | 15.89 / 7.59 / 4.73           | 282.9 / 185.7 / 132.5            | 58.5 / 97.2 / 138.9               |
> | Ours (DWT)        |    9.48 /  4.20 / 4.40           | 195.6 / 117.3 / 84.8       |  78.8 / 136.9 / 199.2     |
> | Ours (DCT)        | 9.60 / 4.19 / 4.37            | 194.4 / 115.1 / 82.5            | 78.6 / 138.3 / 201.2           |
> | Ours (Laplacian)  | 9.77 / 4.28 / 4.28            | 200.6 / 120.0 / 85.8             | 77.9 / 135.6 / 196.8              |
>
> **Table R2:** Quantitative comparison of DFM with CogVideo AE, equipped with various decomposition methods, and trained on ImageNet 512px with a DiT-XL/2 backbone. Numbers correspond to CFG values of 1.0, 1.25 and 1.5.
>
> | Experiment        | FID_50k (1.0 / 1.25 / 1.5)    | FDD_50k (1.0 / 1.25 / 1.5)       | IS (1.0 / 1.25 / 1.5)             |
> |------------------|-------------------------------|----------------------------------|-----------------------------------|
> | Flow Matching     | 16.94 / 8.59 / 5.84           | 319.8 / 218.4 / 159.5         | 58.5 / 96.0 / 137.3            |
> | Ours (DWT)        |   8.72 / 4.26 / 5.08             | 206.9 / 124.5 / 89.2           |  92.3 / 159.2 / 223.5        |
> | Ours (DCT)        | 8.75 / 4.33 / 5.18            | 207.5 / 125.8 / 90.0          | 92.5 / 158.4 / 223.0             |
> | Ours (Laplacian)  | 8.80 / 4.36 / 5.14            | 208.4 / 125.7 / 90.2          | 91.7 / 158.8 / 222.0           |
>
> **Question 2**
> > The paper claims that the progressive modeling achieves better generation performance than Flow Matching. Does this improvement come with greater complexity for training and inference?
>
> While our proposed method maintains **identical training compute** as the baselines, it indeed introduces additional components compared to single-stage models such as Flow Matching. However, some added complexity is an inherent trade-off in progressive modeling. Previous progressive generators improve the performance over the single-scale models but introduce a large degree of complexity such as using multiple models (e.g. Cascaded Models), or incorporating renoising tricks as in Pyramidal Flow (See L. 33-39). A main design principle of DFM is to improve the benefits of progressive models while reducing such complexities by using a **single model, and a training objective and denoising process** closely following single-scale methods. The main additional components introduced by DFM are limited to (i) lightweight decomposition step with an off-the-shelf method (e.g. Laplacian), (ii) sampling multiple timesteps (rather than one), (iii) additional lightweight patchification layers and (iii) maintaining extra hyper-parameters. In practice, these changes are concise and introduce only a marginal time overhead of 1.2% in a single training iteration. We have provided in Appendix A.5 concrete guidance on setting hyperparameters and showed in the paper that following these choices, DFM consistently achieves superior performance on a wide range of generative tasks including image and video generation, and large scale text-to-Image synthesis. We also demonstrated that, under identical hyper-parameters, our method is generalizable to different resolutions (see Table. 2 of the main paper) and different autoencoder and decomposition methods (see our response to Question 1).
>
> **Question 3**
> > During training, do you sample time steps for different scales jointly? Does that need a lot of samples to approximate the $p_t$ expectation?
>
> DFM operates on different stages, where each stage contains multi-scale representations. For instance, for the first stage modeling, only the first (low-frequency) scale is modeled, while in the second stage, both scales (low and high frequency) are jointly modeled. During training, we first sample the stage number from a discrete probability distribution (L. 158). Then, we sample the timesteps of different scales **independently** (L.140), where each scale’s timestep is drawn from a **predetermined** logit-normal distribution, parametrized by a location (m_s), where s is the scale index within the sampled stage. We ablated over the choice of m_s in Table. 2 (b), and found that it is beneficial to have the distribution of the **last scale** to use location (m_s=0), aligning with common practice in flow matching [1-4], while shifting the location of **earlier scales** toward lower noisy levels (higher m_s value). This better simulates the inference-time behavior where during the generation of scale s, the previous scale s−1 has already undergone significant denoising from previous stages. Please see Section 4.2 ("Training timestep distribution") for more details.
>
> To clarify this, we have revised Eq. (4) as follows:
>
> $$
> \\mathcal{L}
>   = E_{Q\\sim p_{q},\\; t^{1}\\sim p_{t, Q}^{1},\\dots,t^{S}\\sim p_{t, Q}^{S},\\; X_{1}\\sim p_{d},\\; X_{0}\\sim p_{n}}
>     \\sum_{s=1}^{S}
>       M^{s}\\,
>       \\left\\lVert
>         \\mathcal{G}\\bigl(P_{t^{1},\\dots,t^{S}},\\,t^{1},\\dots,t^{S}\\bigr)
>         - v^{s}
>       \\right\\rVert_{2}^{2}\\,.
> $$
>
> Here, we introduced a new notation Q that corresponds to the stage number (corresponding to the number of modeled scales) for the current training iteration, and revised the discrete distribution notation to $p_q$. $p_{t, Q}^s$ denotes the timestep distribution of the scale s within stage Q. In DFM, for the two scales case ($Q=2$), $p_{t,Q}^1$ follows a logit-normal distribution with location $m_{1} = 0 $ and $p_{t,Q}^0$ follows a logit-normal distribution with location $m_{0} = 1.5$.
>
> Below, we include a pseudo code of the time step sampling, which we will include in the paper for better clarity.
>
> $$
> \\begin{array}{l}
> \\mathrm{procedure}\\;SampleTrainingTimesteps(S):\\\\
> \\mathrm{inputs:}\\;S \\text{    ; total number of scales}\\\\
> \\mathrm{outputs:}\\;(t^{1},\\ldots,t^{S})\\\\[6pt]
> Q\\leftarrow\\mathrm{SampleDiscreteDistribution}(p_{q}) \\text{ ; stage index corresponding to number of modeled scales}\\\\
> t^{Q}\\leftarrow\\mathrm{SampleLogitNormal}(m=0)\\\\
> \\mathrm{for}\\;s=1\\;\\mathrm{to}\\;Q-1:\\\\
> \\quad t^{s}\\leftarrow\\mathrm{SampleLogitNormal}(m=1.5)\\\\
> \\mathrm{end\\;for}\\\\
> \\mathrm{for}\\;s=Q+1\\;\\mathrm{to}\\;S:\\\\
> \\quad t^{s}\\leftarrow 0\\\\
> \\mathrm{end\\;for}\\\\
> \\mathrm{return}\\;(t^{1},\\ldots,t^{S})
> \\end{array}
> $$
>
> We have revised the method section to better distinguish between “stage” and “scale” and clarify the timestep sampling strategy. We hope that this answers the reviewer’s concern and we remain available to clarify further if needed.
>
> **Question 4**
> > Typos: add-hoc samplers (in Abstract), and the formulate the .... (line 140), etc.
>
> We thank the reviewer for finding the typos. We corrected them in our revised version and performed an additional pass of proofreading.
>
> **References**
>
> [1] Scaling Rectified Flow Transformers for High-Resolution Image Synthesis, ICML 2024
>
> [2] FasterDiT: Towards Faster Diffusion Transformers Training without Architecture Modification, NeurIPS 2024
>
> [3] SnapGen: Taming High-Resolution Text-to-Image Models for Mobile Devices with Efficient Architectures and Training, CVPR 2025
>
> [4] Wan: Open and Advanced Large-Scale Video Generative Models, 2025

---

> > ### Comment · Reviewer_hhY3 · 2025-08-06
> >
> > Thank you for your rebuttal. The additional experimental results provided by the authors are appreciated. I maintain my rating and lean toward acceptance.

---

### Note · Authors · 2025-08-13

We sincerely thank the reviewers for their constructive feedback and comments.

We are pleased that our work received an overall positive assessment before the discussion period, and that all reviewers confirmed their outstanding concerns were resolved afterward, recommending acceptance of our paper. Should the paper be accepted, we will revise the final version to incorporate the reviewers feedback.

---

### Decision · Program_Chairs · 2025-09-17

**Decision:**

Accept (poster)

**Comment:**

This paper proposed a progressive generative modeling framework, termed Decomposable Flow Matching (DFM), built atop flow matching techniques. The core idea is as follows: DFM decomposes input images (or videos) into multi-scale representations using user-defined strategies (e.g., Laplacian pyramid, DWT, DCT, Fourier transforms). Flow matching is then applied independently at each scale, allowing the model to learn and generate at different resolutions. During sampling, the model generates images progressively, traversing from coarse to fine scales. The authors claim that this approach avoids the need for expert-designed models or ad hoc samplers. The authors also show experimentally that DFM achieves significant improvements in image and video generation quality.

The paper has two main strengths:

- DFM is conceptually simple, easy to implement, and agnostic to the choice of decomposition. As a result, DFM could be widely applicable.
- DFM has significant performance gains over state-of-the-art methods on multiple benchmarks, with comprehensive ablations and analysis of hyperparameters. The evaluation is thorough, covering various tasks, and includes analysis of critical design choices (e.g., time-step distributions, scale selection). The approach could make cascaded diffusion models obsolete by offering a cleaner, more effective alternative.

The paper has the following weaknesses:

- The review team noted that the paper is not well written, with unclear explanations of key design choices, including the rationale for the logit-normal distribution for time-step sampling, and the details of the multi-scale decomposition. The paper also suffers from typos and formatting issues that detract from the paper’s professionalism.
- The paper lacks a principled approach for selecting the number and size of scales, and it shifts the difficulty of choosing these parameters to the users.
- There are concerns about fairness in baseline comparisons, particularly regarding the use of scale-equivariant autoencoders in the proposed method but not in baselines. Some ablations are missing, such as the effect of using a uniform distribution for time-step sampling.
- While the method is original in its unified approach, some reviewers feel the novelty is incremental, as progressive modeling and multi-scale decompositions are established ideas.

Overall, the review team has a positive assessment of the paper (3 weak accept, 1 accept). The consensus is that DFM is a clear step forward in simplifying and improving progressive generative modeling, and that the empirical results are strong and well-supported.

As a final note, I encourage the authors to revise the paper for clarity, provide more principled guidance on decomposition choices, and ensure fair baseline comparisons. With these improvements, the work could have a broader impact.